# Dynamics of bacterial population growth in biofilms resemble spatial and structural aspects of urbanization

Amauri J. Paula[1,2 ✉], Geelsu Hwang [2,3,4 ✉] & Hyun Koo [2,4 ✉]

Biofilms develop from bacteria bound on surfaces that grow into structured communities (microcolonies). Although surface topography is known to affect bacterial colonization, how multiple individual settlers develop into microcolonies simultaneously remains under-explored. Here, we use multiscale population-growth and 3D-morphometric analyses to assess the spatiotemporal development of hundreds of bacterial colonizers towards submillimeter-scale microcolony communities. Using an oral bacterium (*Streptococcus mutans*), we find that microbial cells settle on the surface randomly under sucrose-rich conditions, regardless of surface topography. However, only a subset of colonizers display clustering behavior and growth following a power law. These active colonizers expand three-dimensionally by amalgamating neighboring bacteria into densely populated microcolonies. Clustering and microcolony assembly are dependent on exopolysaccharides, while population growth dynamics and spatial structure are affected by cooperative or antagonistic microbes. Our work suggests that biofilm assembly resembles certain spatial-structural features of urbanization, where population growth and expansion can be influenced by type of settlers, neighboring cells, and further community merging and scaffolding occurring at various scales.

[1] Solid-Biological Interface Group (SolBIN), Department of Physics, Universidade Federal do Ceará, P.O. Box 6030, 60455-900 Fortaleza, CE, Brazil. [2] Biofilm Research Labs, Levy Center for Oral Health, Department of Orthodontics, Divisions of Pediatric Dentistry and Community Oral Health, School of Dental Medicine, University of Pennsylvania, Philadelphia, PA 19104, USA. [3] Department of Preventive and Restorative Sciences, School of Dental Medicine, University of Pennsylvania, Pennsylvania, PA, USA. [4] Center for Innovation & Precision Dentistry, School of Dental Medicine, University of Pennsylvania, Philadelphia, PA 19104, USA. ✉email: amaurijp@gmail.com; geelsuh@upenn.edu; koohy@upenn.edu

Previous studies have shown that the development of biofilms is a dynamic process that involves bacterial adhesion and clustering, and further structural organization into large communities (e.g., microcolonies)[1–5]. Further findings also revealed the importance of short-range (cell–cell or cell–matrix) interactions for the development of structured communities at a single-microcolony level[6–12]. Specifically, the three-dimensional evolution of *V. cholera* biofilm community was found to be associated with multiple factors such as cell orientation, cell density, and cell–cell distance or distance between two microcolonies[6,8]. However, a question arises on how the initial colonizers spatially distributed at different locations on the surface grow and subsequently develop into microcolony communities. Mathematical modeling has been employed to assess multiple variables simultaneously to predict bacterial growth dynamics, providing important theoretical principles of the biofilm development process. For example, growth laws have been applied to assess the complexity of cell physiology regulation and bacterial growth dynamics using simple mathematical relations as quantitative models[13–18]. Yet, prediction of the biofilm development pattern at larger-scales remains unresolved due to intrinsic heterogeneity of bacterial colonization, spatial localization, and growth behavior[1,19–21], which cannot be captured at small length and short-term scales. To our knowledge, the growth laws of an entire population of surface-colonizing cells from initial clusters to structured biofilm communities across multiple locations, time, and length scales have not been determined experimentally.

To achieve this, we perform multiscale biofilm growth monitoring, from single-cell to multicellular communities, by employing high-resolution time-lapsed confocal imaging and microfluidics combined with a 3D scalar mapping algorithm to assess the spatial and structural aspects of population growth and expansion. Given the spatial heterogeneity of the location of initial colonizers, we assess the surface topography and the bacterial colonization profile of *Streptococcus mutans* (a biofilm-forming model organism), and the subsequent growth dynamics of these initial colonizers at multiple-length scales and dimensions either alone or in mixed species. We track the entire population of surface colonizers (C) individually and collectively throughout the biofilm development period. A large population of initial colonizers is analyzed (C > 800) to characterize the spatial heterogeneities and contributions of bacterial binding location, growth law coefficients, and the role of EPS matrices. Furthermore, the influence of an antagonist (*Streptococcus oralis*) or a symbiotic (*Candida albicans*) organism on *S. mutans* growth dynamics is assessed using this approach. The data reveal how individual settlers (initial colonizers) are spatially distributed across the substratum and develop three-dimensionally into structured multicellular communities as a function of matrix-scaffolding and the type of neighboring cells (antagonistic or symbiotic). The growth dynamics display spatial and structural patterns similar to urbanization, whereby some settlers stayed static while others grow into aggregates (villages) that further expand into densely populated microcolonies enclosed to well-defined boundaries (cities), which in turn merge to each other resulting in a larger biofilm superstructure (megacity). This conceptual framework may lead to alternative ways of studying the biofilm-assembly mechanisms and assessing therapeutic strategies that include events at large scale.

## Results

### Biofilm spatiotemporal population analysis (BioSPA).
The evolution of a biofilm community from single-cell to multicellular level was assessed by employing an analytical tool, designed for spatiotemporal analysis at multi-length scale across three dimensions, to track multiple cells and clusters regarding growth dynamics, morphology (size/shape), and surface topography simultaneously (termed Biofilm Spatiotemporal Population Analysis or BioSPA, see "Methods", Supplementary Fig. 1). Several analyses can be performed, including (i) identification of spatial patterns of bacterial binding according to surface topographies (the terrain); (ii) assessment of the growth of the entire population of colonizing microorganisms (the individuals) on a given surface; (iii) visualization of how they organize themselves into structured communities (the habitation); and (iv) evaluation of symbiotic/competition behavior among the population (the residents interactions). The BioSPA and a tutorial guide are readily available in GitHub (https://github.com/amaurijp/BioSPA).

### Bacterial colonization on an apatitic surface.
Biofilm-assembly process involves bacterial colonization and further growth of the colonizers on the surface, leading to structured microbial communities[1]. Given the importance of surface topography on bacterial colonization, we first assessed the spatial distribution of initial colonizers on the hydroxyapatite disc (HAD) surface. We employed a hybrid confocal imaging-surface topography approach for the simultaneous analysis of colonizing bacterial structures and HAD surface topography from submicron to submillimeter length scales (Fig. 1). The bacterial cells were first incubated with HAD disc surface for 60 min in static condition to allow bacterial binding and then aseptically transferred to a flow chamber. Before starting the flow, the population of surface colonizers (C), grouped based on the number of adherent bacterial cells, was examined using our imaging approach (Fig. 1a–c). The bacteria channel revealed a large population of surface colonizers (C): C > 200 in an area of $319.45 \times 319.45\ \mu m$ with a varied number of bacterial cells ($P_0$). Colonizers with $1 \leq P_0 \leq 5$ (single cells) represented 60–70% of C, clusters ($5 < P_0 \leq 50$) were 25–35%, and aggregates ($50 < P_0 \leq 300$) were 1–6%. By overlapping the binarized Z-projection of the bacterial channel on the image stack of the HAD surface topography, we identified all sites upon which bacterial surface colonizers were attached ($t_0$) (blue color in Fig. 2a; see arrows). Topographical parameters such as average roughness ($S_a$), root-mean-square roughness ($S_q$), and skewness ($S_{sk}$) were then calculated for each bacterial colonization site. Since several experiments were performed independently, a large number of surface colonizers were analyzed (C > 800; Fig. 2b–g).

Traditionally, surface sensing and attachment mechanisms of microorganisms have been characterized by considering the substratum as a flat surface, thus overlooking topographical heterogeneity[22–24]. To address this, we locally analyzed the surface topography of the sites upon which bacterial colonizers were bound. This was performed by cropping the image stack containing the topographical profile (heighmap) exactly at the regions where bacterial colonizers were found at $t_0$ and by calculating the surface roughness parameters $S_a$, $S_q$, and $S_{sk}$ for these sites. Furthermore, we also measured $S_a$, $S_q$, and $S_{sk}$ for the entire surface profile by splitting the heightmap ($319.57\ \mu m \times 319.57\ \mu m$) into squared areas of $\Delta L \times \Delta L$ ($3–115\ \mu m^2$), termed multiscale area mapping (MAS), to calculate the roughness of all possible binding sites on the surface, in all possible scales (Fig. 2a). We found that bacteria initially bound ($t_0$) to sites with low $S_a$ and $S_q$ values (<2.5 μm), regardless of the occupied area. $S_{sk}$ values for colonized sites were also symmetrically distributed, indicating that the colonizers adhered to both "hills" and "valleys" present on the HAD. This result suggests flexibility in *S. mutans* mechano-sensing or surface interaction via EPS[25,26], although additional studies are required to further understand their mechanisms of adhesion at the single-cell level[27]. In regard to a possible binding preference on the available surface sites ($S_a = 0.2–7.0\ \mu m$), normalized histograms

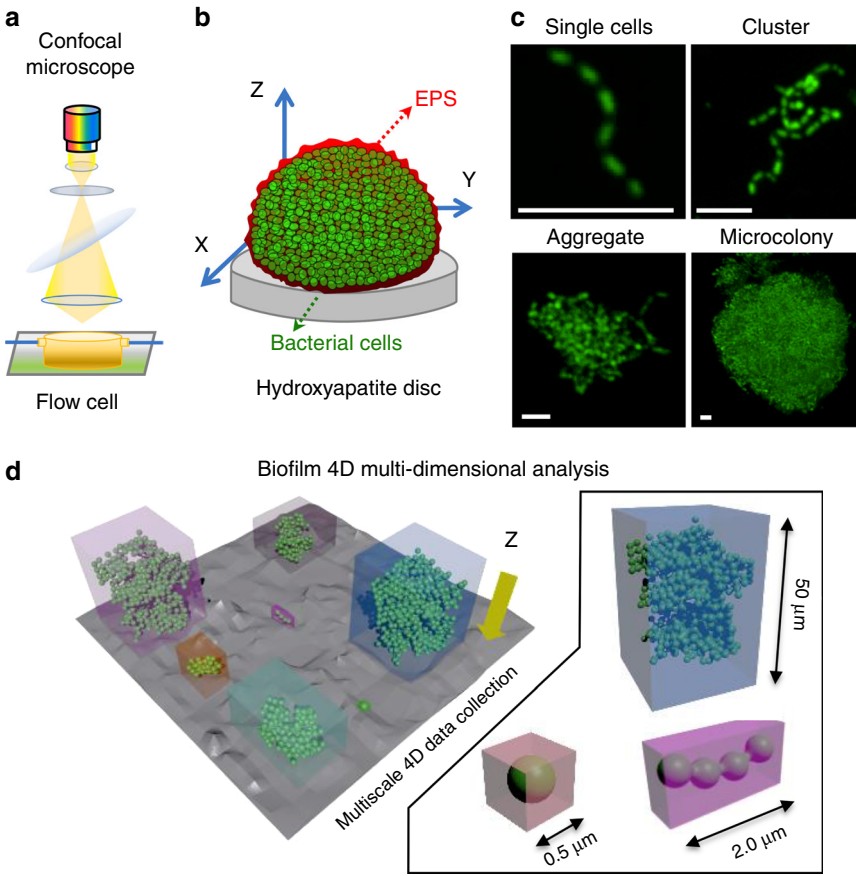

**Fig. 1 Setup of the confocal laser scanning microscope (CLSM) and the image-processing software (BioSPA) to perform the spatial population-growth analysis over time. a** Experimental setup for analyzing microbial colonization and further growth under flow in situ using confocal laser scanning microscopy-surface topography imaging approach. **b** Schematics of biofilm constituents detected by independent signal detection of bacteria (green), exopolysaccharides substances (EPS; red), and hydroxyapatite disc (HAD) surface (gray). The HAD is placed inside the flow cell in the position as shown in the diagram. **c** Representative Z-projection (max intensity) of the CLSM images depicting the evolution of a surface colonizer (single cells, cluster, aggregate) into structured communities (microcolony). White bars: 5 μm. **d** Schematics of the image-processing method to capture the population units (i.e., single cells, clusters, aggregates, and microcolonies) across time and space, i.e., three dimensions and at multiple scales. The data were analyzed using biofilm spatiotemporal population analysis (BioSPA) software.

of $S_a$, $S_q$, and $S_{sk}$ values for non-colonized areas (n.c.a.) overlap those calculated for surface sites containing colonizers at $t_0$ (first column in Fig. 2e–g). This results revealed randomness for the initial attachment of the colonizers.

Upon initiating the flow in the chamber, ~40% of colonizing cells (C) detached, suggesting varying mechanical stability on the surface under fluid shear forces (Supplementary Fig. 2). This detachment occurred in a random fashion, as we observed similar patterns of normalized histograms of $S_a$, $S_q$, and $S_{sk}$ values calculated before ($t_0$) and after initiation of fluid flow (second column in Fig. 2e–g). Once the flow was initiated, the bacterial detachment was abundant across the HAD surface, while the detachment process was significantly decreased after 30 min of flow. Numerous dark red-colored boxes in Supplementary Fig. 2 indicated major detachment events occurred within 30 min. Highlighted boxes in Supplementary Fig. 2 with yellow color showed that detachment of surface colonizers with small sizes (cells and chains) occurred even at later stages of growth (>300 min), but cell clusters remained stably attached. These stably attached colonizers were followed individually to analyze changes in size and shape over time.

**Growth dynamics of surface colonizers.** To investigate the growth dynamics of surface-attached bacterial cells, we tracked

each surface colonizer individually in real-time using 4D scalar field generated from the time-lapsed CLSM stacks (Fig. 3a). Cell growth and further biomass changes were observed where the initial colonizers remained on the HAD under flow (Fig. 3b; population highlighted in colored boxes from dark red to yellow). Histogram with the initial volume values (V(0 min)) for surface colonizers (at $t_0$) and volume values at 420 min (V(420 min)) showed that majority of colonizers did not grow, while colonizers following power law (i.e., V(420 min) > 5 V (0 min)) evolved into various sizes of microcolonies over time (up to 750 μm³) (Fig. 3c, d).

We unexpectedly found that only a subset of all settlers evolved into microcolonies. Thus, two distinctive phenotypes were identified: Static Colonizers (SC), comprising 60% of the initial colonizers, whereby the bacteria bound to the apatitic surface maintained their volume relatively constant (V(t)≈V(0 min)) (Fig. 4a); and dynamic colonizers (DC), the remaining fraction of colonizers exhibited active growth, which expanded their volume by following the relation $V(t) = V(0\ min) + a \cdot t^b$, where "a" is a normalization constant and "b" is the exponential constant of the growth law (Fig. 4c, d). For DC population (V(420 min) » V(0 min)), the growth law becomes the power law[24,28] $V(t) = a \cdot t^b$ ("$t$" in minutes and "V(t)" in μm³), which can precisely model the spatial population-growth dynamics. The influence of the bacterial cell number of the surface colonizers ($P_0$) on the growth

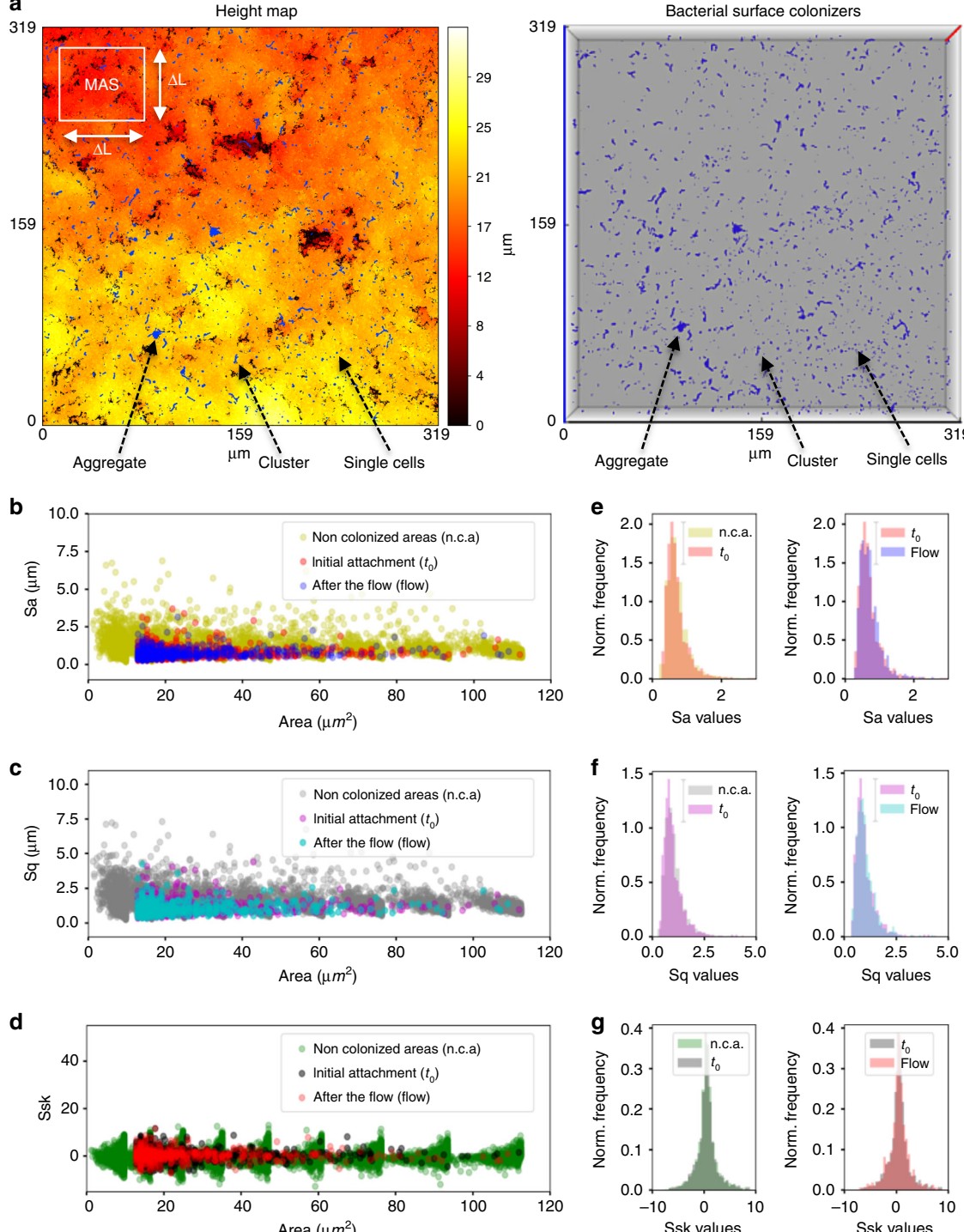

**Fig. 2 Attachment and stability of the bacterial colonizers as a function of the surface topography. a** In situ visualization of surface colonizers (at $t_0$) on hydroxyapatite discs (HAD), overlapped with the surface topography (heightmap, left panel). Surface colonizers are shown in blue in the top view of a 3D representation of the bacterial cells on HAD (right panel). Z-axis (red line in right panel) represents a scale of 48 μm. Initial colonization of the surface consisted of single cells, clusters, and aggregates. **b–d** Topographical parameters average roughness ($S_a$), root-mean-square roughness ($S_q$), and skewness ($S_{sk}$) were determined at $t_0$, and after the nutrient flow started in the chamber (flow). **e–g** Normalized histograms of $S_a$, $S_q$, and $S_{sk}$ values. For non-colonized areas (n.c.a.), the parameters were calculated for the whole-scanned area and at different length scales (multiscale area selection—MAS; shown in **a**). The results represent a large population of bacterial surface colonizers (C > 800), captured from independent experiments ($n = 3$). Source data are provided as a Source Data file.

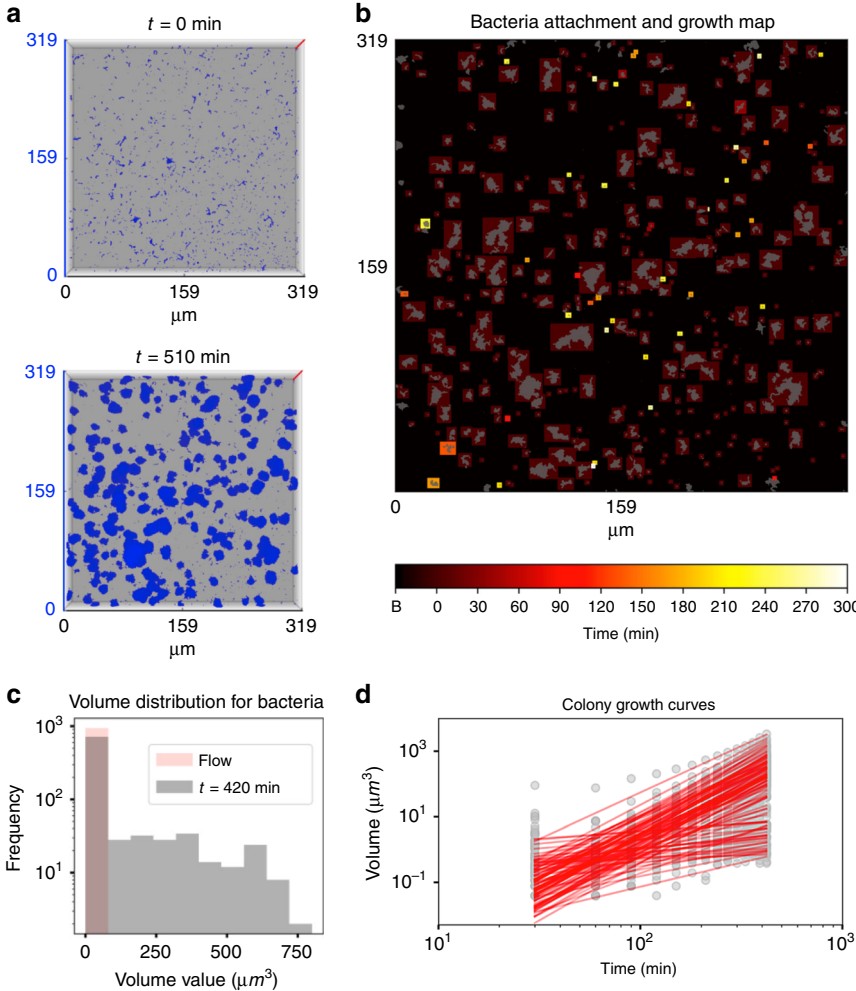

**Fig. 3 Dynamics of spatial population growth of surface-attached bacteria. a** Three-dimensional representation of the *S. mutans* growth under the flow of 1%-w/v sucrose at two different time points (0 and 510 min). Z-axis (red lines in (**a**)) represents a scale of 48 μm. **b** Time-lapsed heatmap of bacteria attachment and growth. Colored boxes indicate the time and location where bacteria attached. Elements shown in gray represent a projection of the bacteria signal at $t = 300$ min. Bacteria without colored boxes were not included in the population analysis. **c** Histogram with the initial volume values (V(0 min)) for surface colonizers at $t_0$ (in pink) and volume values at 420 min (V(420 min); in gray). Flow indicates the time that the culture medium starts to flow. **d** The evolution of the colonizers into microcolonies was assessed by biofilm growth curves (in gray-colored data points for each time point) and fitted curves (red-colored continuous lines) ($n = 3$). Source data are provided as a Source Data file.

dynamics is shown in Fig. 4e–g. The median value between different experiments for the normalization constant $a$ is $1 \times 10^{-5}$, while for exponent $b$ is 2.6 (red lines in Fig. 4f, g). We compared the number of cells of initial colonizers (single cells, clusters, and aggregates) and their biovolumes at 0 min and at the end of the experimental period at 420 min. As shown in Fig. 4a (three colored lines), most of the single cells had less than the volume of 10 μm³ at 420 min (white line), while clusters developed to a structure with a range of $10^2$–$10^3$ μm³ (blue line), and aggregates became large microcolonies (~$10^3$ μm³) (green line). In the static population (SC) (red rectangle in Fig. 4b), however, surface colonizers kept their volumes approximately constant regardless of the initial volume or cell number. Based on this analysis, we classified three biofilm developmental stages for DC population: (i) initial colonization: initial colonizers are spatially heterogeneously distributed on the HAD surface with a varied number of cells ($P(t_0) = P_0$), ranging from single cells ($1 < P_0 \leq 5$) and clusters ($5 < P_0 \leq 50$) to small aggregates ($50 < P_0 \leq 300$); (ii) individual development: certain colonizers (DC) grow and assemble into microcolonies with varying cell density with well-defined shapes and boundaries ($P(t) > 300$); and (iii) mutual development:

subsequent merging of multiple microcolonies into large biofilm structures. Thus, from a spatial and structural perspective, the growth dynamics display a pattern similar to urbanization, whereby some settlers (initial colonizers) stay static while others grow into clusters and aggregates (villages) that further expand toward densely packed microcolonies (cities), which in turn merge to each other resulting in larger biofilm superstructures (megacities).

**Community merging toward a biofilm superstructure**. To understand the physical and biological interactions between bacterial communities, we investigated the possible alterations in population-growth dynamics for multiple communities that underwent a merging process (Fig. 5a, b). We considered the centroids of adjacent communities and used the convex hulls from their contact planes to generate 3D scalar fields, in which the volume occupied by the cells was determined over time (see the Z-projections of the 3D scalar fields in blue, red and yellow in Fig. 5c). Interestingly, we did not find significant changes of the growth rate for the microcolonies during

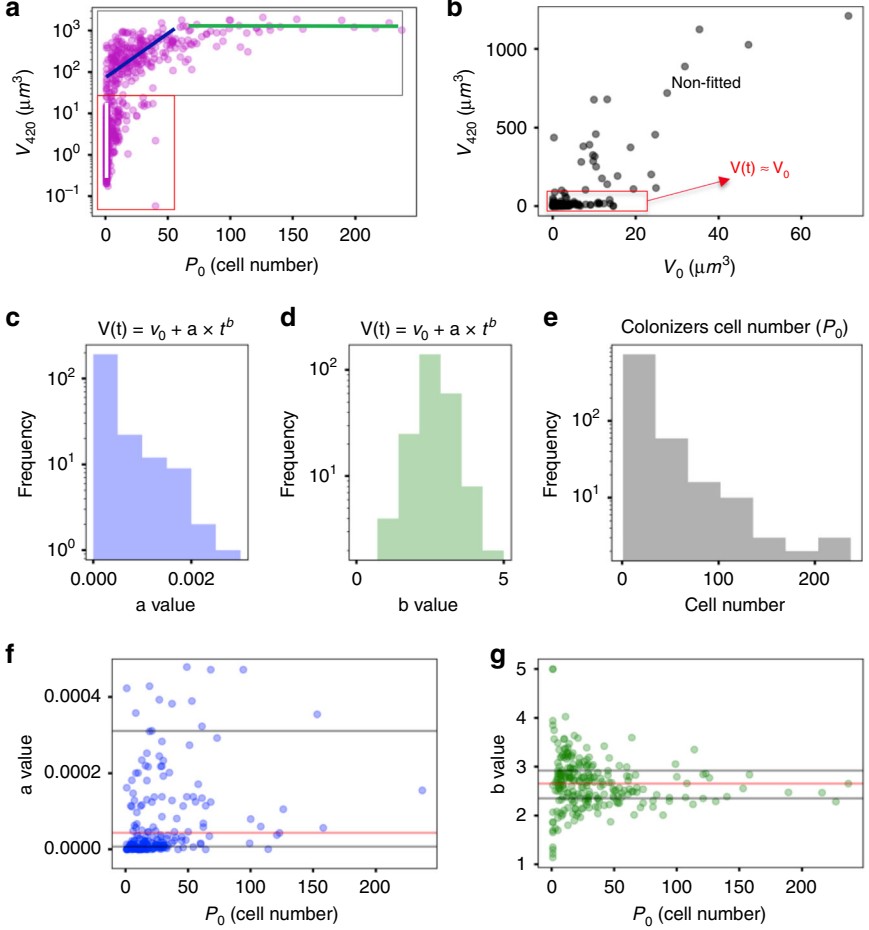

**Fig. 4 Multidimensional population-growth analysis of colonizers during biofilm development. a** V(420) as a function of $P_0$ for all bacterial population units (fitted and non-fitted). Bacterial growth followed the power law $V(t) = a \cdot t^b$ (for V(420 min) »V(0 min)), which represented around 40% of C (dark gray rectangle), termed dynamic colonizers (DC). **b** Distribution of V(420) as a function of V(0) for the non-fitted population and also for surface colonizers that do not grow ($V(t) \approx V(0)$). Non-fitted curves represented less than 0.1% of C. The results represent a large population of bacterial surface colonizers (C > 800), captured from independent experiments ($n = 3$). **c**, **d** Distribution values of the normalization constant ($a$) and the exponent ("$b$") for the fitted curves. **e** Histogram depicting bacterial cell number of the surface colonizers at $t_0$ ($P_0$). **f**, **g** Distribution of "$a$" and "$b$" values as a function of $P_0$. Median is represented with red lines, and 1st and 3rd quartiles with dark gray lines ($n = 3$). Source data are provided as a Source Data file.

merging events under nutrient-rich (1% sucrose) condition (colored vertical lines in Fig. 5d, e indicate the merging moment). Hence, *S. mutans* communities manifest no clear impairment on each other, which collectively evolves into a densely packed biofilm superstructure as a result of several of these merging processes occurring over time.

We next examined whether this merging behavior would change in nutrient-limiting conditions. Thus, we monitored the merging events in low sucrose (0.1%) and in feast and famine cycles (alternating between high and low sucrose; see "Methods"). Under low sucrose condition (0.1% sucrose), the biofilm development was substantially delayed disrupting both microcolony formation and the merging dynamics (retarding the merging events by >3 h vs 1% sucrose). Under feast and famine cycles, we found that the growth rate curves oscillate accordingly, but the oscillation was the same for all monitored communities undergoing merging, which indicates that even in nutrients scarcity *S. mutans* biofilm communities do not manifest a "prey" behavior on each other, i.e., one grows while the other halts or shrinks. Although competition between communities cannot be excluded, its existence in *S. mutans* biofilms does not impair the development of the population as a whole.

**Role of EPS on bacterial growth dynamics and biofilm architecture**. The production of extracellular polymeric substances such as exopolysaccharides (EPS) has been considered a vital attribute for the biofilm lifecycle[21,29]. We assessed whether the evolution of bacterial colonizers toward structured communities (microcolonies) is directly associated with EPS. EPS (glucans) are produced by *S. mutans* via exoenzymes termed glucosyltransferases (Gtf) using sucrose as substrate[30]. Using glucan-specific labeling technique based on Gtf activity[30], we were able to assess spatiotemporal production of EPS in situ during the various stages of DC growth. The location of glucans was determined as the bacterial cells mediate intercellular adhesion, clustering and microcolony development from submicron to tens of micron in scale.

To investigate the role of EPS during the entire development process, we employed a two-pronged strategy: substrate and enzymatic-based approaches. First, we replaced sucrose with an equimolar concentration of glucose and fructose (sucrose monosaccharide moieties). Glucose and fructose cannot be utilized by Gtfs to produce EPS glucans yet they are efficiently metabolized by *S. mutans* allowing similar growth rates compared with sucrose-grown conditions. Interestingly, we observed that, in the presence of glucose and fructose, the colonizers grew as

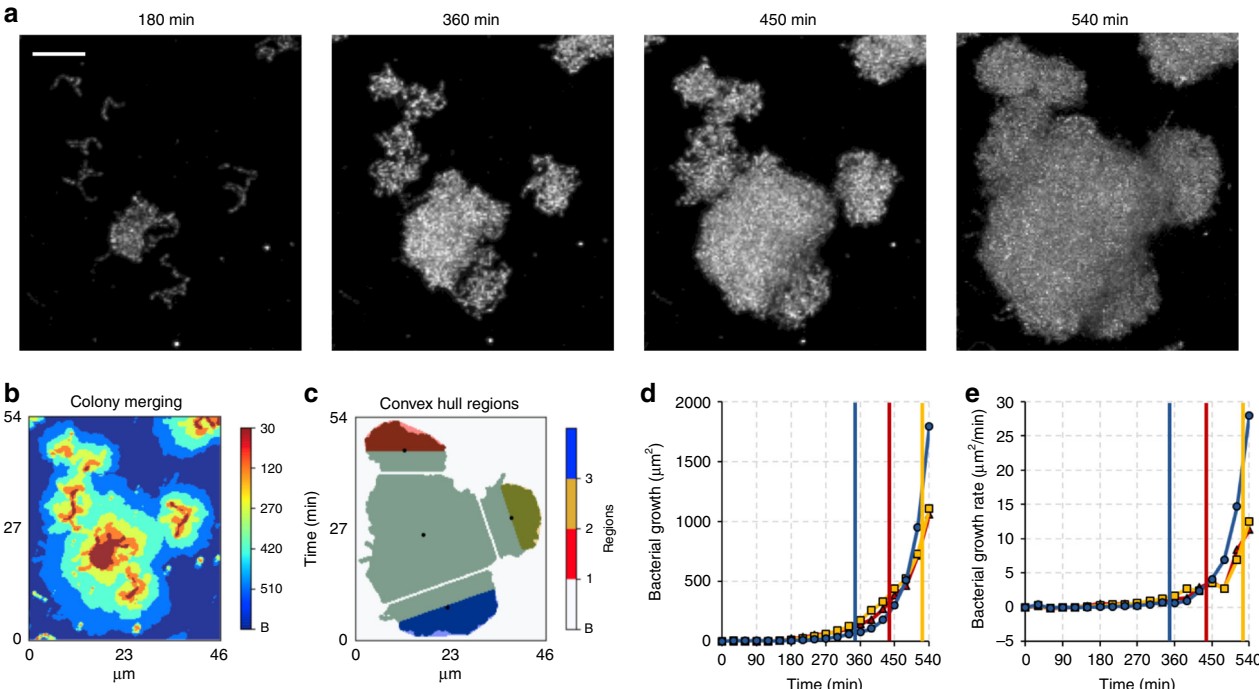

**Fig. 5 Merging behavior of neighboring microcolonies. a** Time-lapsed confocal images of microcolonies development. Scale bars indicate 10 μm.
**b** Projection map of binarized stacks from 540 min showing elements merging of multiple *S. mutans* microcolonies along time. **c** 3D scalar-field analysis
(represented as *Z*-projection) performed for each marginal element merging to the central one. The volumes were calculated for three colored regions of
the marginal elements as shown in **c**. Black dots represent the centroids of each element at 540 min, determined on the *Z*-projection. **d** Growth and
**e** growth rate curve of the respective marginal elements. Colored lines indicate the moment of merging to the central element ($n = 3$). Source data are
provided as a Source Data file.

interwoven chains, forming a filamentous mesh-like morphology
(Fig. 6a) without detectable EPS or development of microcolo-
nies. In sharp contrast, the EPS were produced in the presence of
sucrose and found co-localized or closely associated on the
surface and in between individual cells. The EPS also enmeshed
the cells forming a matrix that bridged them together into
clusters, leading to structured aggregates or microcolonies as
the time elapsed (Fig. 6b), resembling the matrix-mediated *V.
cholerae* and *P. aeruginosa* cell clustering[7,12]. To further assess the
role of EPS on the dynamics of bacterial community structuring,
we added EPS-degrading enzymes (mutanase and dextranase)
during biofilm growth. These enzymes can effectively reduce the
accumulation of EPS by digesting them without affecting bacterial
viability or growth rate[31–33], thus allowing assessment of matrix
contributions to biofilm development. As shown in Fig. 6c, EPS
was minimally detected throughout the time-lapsed experiment
when glucanohydrolases (mutanase and dextranase) were present
during *S. mutans* biofilm growth in 1% sucrose, indicating its
degradation by the enzymes during the experimental period. As a
result, the cells were unable to form structured microcolonies,
resembling the morphology of those grown in glucose and
fructose (Fig. 6a).

To quantitatively compare the effect of the EPS production on
the morphological evolution of DC, we determined the volume of
the convex hull[34] (qhull Volume) for each microorganism
community captured in the stack (see Supplementary Fig. 3).
By calculating the ratio of the convex hull volume over total
biomass volume (qhull volume/volume), we determined the
pattern of each element of the population, and we quantitatively
determined how similar the community morphology is to a
sphere or to a branched pattern (i.e., determination of the solidity;
Supplementary Fig. 3) in each experimental condition. When the
biofilm was grown in 1% sucrose, most of the elements in the

population had an $R_{(qhullVolume/Volume)}$ smaller than 4, and this
ratio decreases at later periods when the element volume
increases (around $R_{(qhullVolume/Volume)} = 2$ at 360 min). On the
other hand, in the presence of 0.5% glucose + 0.5% fructose, the
communities evolve to a high $R_{(qhullVolume/Volume)}$ (>4), even for
elements with small volumes (<180 μm³). Interestingly, morpho-
logical homogeneity was observed in the shape of the commu-
nities formed in 1% sucrose over time (horizontally scattered
points in Fig. 6d left panel). Furthermore, when mutanase and
dextranase were added, the community structural shape was
similar to that observed with 0.5% glucose + 0.5% fructose, in
which EPS is absent (Fig. 6d right panel). For both conditions,
whereby EPS-matrix assembly was compromised, the bacterial
volume/accumulation was significantly lower compared with
that in 1% sucrose as time elapsed (Fig. 6e left panel). This
decrease in the biovolume reflects on the coefficient "a" value of
the equation that fitted the curves (Fig. 6e left panel), in which the
"a" value for the 1%-Suc experiment (EPS present) is several-fold
larger (>5) than for experiments where EPS is absent (0.5% Fru +
0.5% Glu) or degraded (1% Suc + Dex/Mut). The same behavior
was observed when the surface occupation by bacteria was
analyzed over time: EPS production favors *S. mutans* to spread
and colonize more efficiently the available surface (Fig. 6e
right panel).

We also determined that the mechanical stability of the
microbial communities (surface attachment) is also associated
with EPS, which prevent the sloughing of clusters and micro-
colonies as they grow under the flow. Indeed, the presence of
EPS-degrading enzymes (mutanase and dextranase) resulted in
the displacement of bacterial clusters on HAD (Supplementary
Fig. 4A). In addition, time-lapsed single-cell imaging analysis
showed that *S. mutans* cell chains grown in 0.5% glucose + 0.5%
fructose (EPS glucans absent) were unstable when oriented

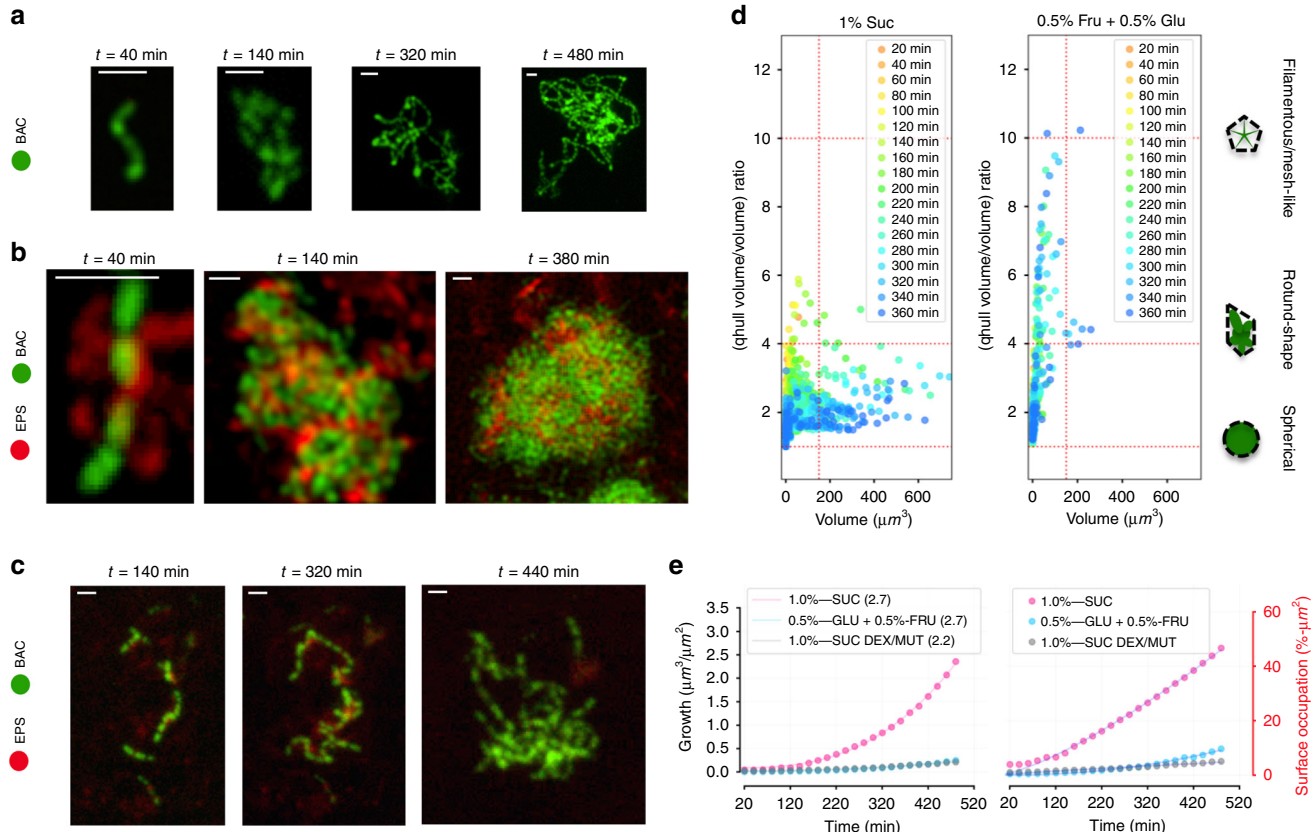

**Fig. 6 EPS production in situ mediates cell aggregation and microcolony structuring.** Z-projection (max intensity) of the CLSM images stack showing *S. mutans* growth on HAD in (**a**) 0.5 %-w/v fructose + 0.5 %-w/v glucose (0.5% Fru + 0.5% Glu) for bacteria (green), in (**b**) 1 %-w/v sucrose (1% Suc) for bacteria (green) and extracellular polymeric matrix (red), and in (**c**) 1 %-w/v sucrose with EPS-degrading enzymes (1% Suc, Dex/Mut) for bacteria (green) and extracellular polymeric matrix (red) over time. Scale bar indicates 2 μm. **d** Morphological analysis of the biofilm-forming elements formed over time, evaluated by the ratio between the volume and the convex hull volume (Volume/qhull volume). **e** Curves showing volume growth and surface area occupation of *S. mutans* growing on HAD in different conditions (n = 3). Source data are provided as a Source Data file.

orthogonally, collapsing to the bottom over time (Supplementary Fig. 4B; see arrows). Altogether, degradation or absence of EPS scaffolding resulted in bacterial communities that lacked structural integrity and stability, disrupting the ability of colonizing cells to pack together and expand tri-dimensionally during growth, which prevented the establishment of organized microcolonies.

**Growth dynamics in mixed communities.** Biofilms found in clinical or environmental settings are often comprised of different species that can co-exist or antagonize like good and bad neighbors. Previous studies have shown an intriguing cross-kingdom symbiotic relationship between the fungus *Candida albicans* and *S. mutans*[35–37], whereby both organisms can grow together in sucrose. In contrast, *Streptococcus oralis* can antagonize by producing hydrogen peroxide ($H_2O_2$)[38], which can inhibit *S. mutans* growth. However, how these relationships locally affect the population-growth dynamics from initial colonizers to further development into biofilms remains unexplored. We found that the growth of *S. mutans* co-cultured with *S. oralis* J22 (Sm-So) was drastically impaired, as determined by the power law (V(t) = a·t^b). Curve fitting analyses showed a > five-fold lower a value compared with that from either single-species *S. mutans* (Sm) or *C. albicans*-*S. mutans* mixed biofilms (Sm-Ca; Fig. 7a). Such changes resulted in *S. oralis* overtaking *S. mutans* population and disrupting further growth. In contrast, the presence of *C.*

*albicans* did not alter the growth dynamics of *S. mutans* (Fig. 7a), while causing major structural changes in the mixed biofilm. The biovolumes of *S. mutans* in mixed Sm-Ca biofilm at early time points (<100 min; highlighted red box) were substantially higher (green dots in Fig. 7a) compared with those from *S. mutans* in single-species biofilm (dark purple dots). This disparity may be associated with changes in the structural organization and/or biovolume when *C. albicans* were co-cultured with *S. mutans* during mixed-species biofilm development.

To further investigate this cross-kingdom interaction, the volume occupation was calculated for both the bacterium and fungus based on each individual fluorescence signal (Fig. 7b, c) by separating four representative 3D regions in the CLSM images stack (R1, R2, R3, and R4). In the four regions, *S. mutans* in the mixed biofilms grew under the same power law previously identified when it developed alone (Volume = a·t^b). Exponent b varied between 2.3 and 2.5 in the regions, indicating that *C. albicans* cell did not compete with *S. mutans* and were able to share the nutrient (1% sucrose), thereby maintaining similar bacterial growth dynamics to that found in single-species *S. mutans* biofilm. However, when assessing the morphological development of the biofilm, we found that *C. albicans* caused intriguing changes in the spatial arrangement of the cells when co-cultured with *S. mutans* on the HAD surface. *C. albicans* grew occupying interstitial spaces of *S. mutans* communities (Figs. 7c–f), "stitching" different clusters and microcolonies together. The chain of yeast cells were bound in-between the *S.*

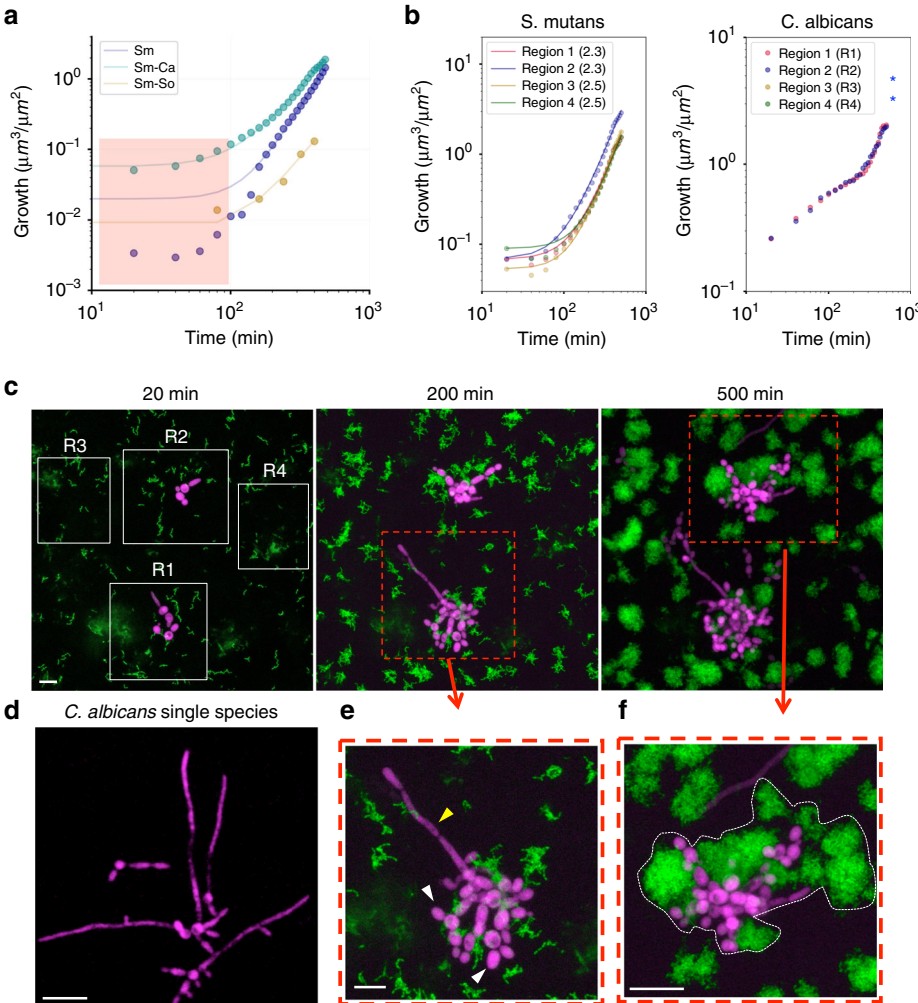

**Fig. 7 Growth dynamics in mixed communities. a** Growth curve (dots) and fitting (lines) for *S. mutans* in single-species biofilm (Sm; dark purple dots), in cross-kingdom biofilm (Sm-Ca; green dots), and in mixed-bacterial species biofilm (Sm-So; yellow dots). **b** Localized *S. mutans* and *C. albicans* growth curves at distinct sites (R1, R2, R3, and R4 as shown in **c**). Exponent *b* values are in parenthesis. **c** Z-projection (max intensity) of the CLSM images stack showing growth of *S. mutans* and *C. albicans* on HAD, with bacteria (green) and fungi (purple) channels, (**e, f**) zoom-in images of *S. mutans* superstructures formed in the presence of *C. albicans* (also highlighted by dotted-white line in **f**). **d** Z-projection of the CLSM image of *C. albicans* single-species biofilms. Scale bars indicate 10 μm. Source data are provided as a Source Data file.

*mutans* clusters even at early time points (at 200 min), leading to larger structures (at 500 min) than the microcolonies without *C. albicans* (i.e., R1 and R2). Interestingly, hyphae formation by *C. albicans* largely decreases when compared with a single-species *C. albicans* biofilm (Fig. 7d). In situ analyses revealed that *C. albicans* in the close proximity to *S. mutans* maintained yeast form (white arrows), while remotely located *C. albicans* displayed hyphal form (yellow arrow; Fig. 7c–i).

Altogether, the data reveal unique structural organization between bacteria and fungi at micron-scale which substantially impacted the morphology at large scale. It appears that the co-existence between *S. mutans* and *C. albicans* can promote the merging events, which resulted in larger mixed communities (possibly with greater stability) at earlier stages of biofilm development. Hence, cooperation between different neighboring communities accelerates the conurbation process, leading to the formation of a larger biofilm superstructure (city) more rapidly.

## Discussion

Biofilms are surface-associated multicellular communities embedded in a matrix of extracellular polymeric substances that

cause many human infections and are broadly implicated in medical and industrial biofouling[39]. The data presented here reveal that biofilm-assembly dynamics involve multiscale events occurring simultaneously across space and time whereby only a subset of initial colonizers at different locations actively grow and organize themselves into densely populated micron-scale micro-colonies that further expand and merge to each other forming submillimeter superstructures. The influence of surface properties such as roughness on initial bacterial attachment has been previously reported[40–43]. Although HAD disc exhibits randomly distributed peaks and valleys across the surface with roughness at μm level, we did not observe a significant association between surface topographical parameters and *S. mutans* initial colonizing pattern under sucrose-rich condition (Fig. 2). One possible explanation may be related to *S. mutans*-secreted glucosyl-transferases that can produce glucans locally using sucrose as a substrate to promote direct bacterial adhesion on HAD via glucan-binding protein on the cell membrane[26], thereby attenuating the influences of the surface topography. Following initial binding, the actively growing colonizers could be fitted with high precision by the power law $V(t) = a \cdot t^{b}$, where volume (V) was

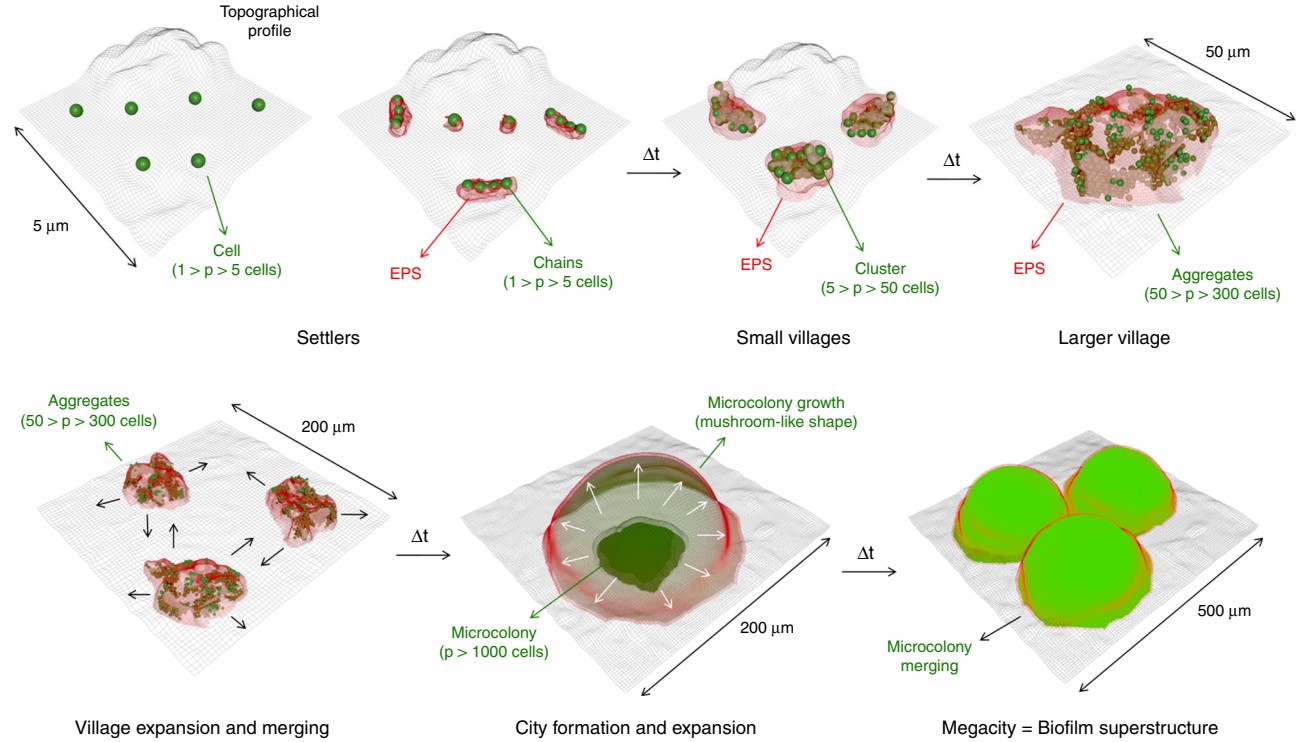

**Fig. 8 Schematic diagram of spatiotemporal population growth during biofilm development.** Dynamics of bacterial population growth during biofilm development resemble spatial and structural aspects of urbanization.

given in μm$^3$ and time ($t$) in minutes. We also found that EPS glucan production by *S. mutans* directly impacts the value of coefficient "a", but not "b". Clustering and spatial structuring were dependent on EPS production in situ that provided an adhesive scaffold to keep the cells together and stably attached on the surface, allowing verticalization and tri-dimensional expansion of the microcolony structure.

Recently, Kragh et al.[44] reported that aggregates of *Pseudomonas aeruginosa* are more competitive than single cells in terms of growing into structured biofilms in a nutrient-rich environment because of their larger access to nutrients along the Z-axis (i.e., aggregates are "taller" than single cells). Interestingly, we found that most of the initial single-cells colonizers ($P_0 < 5$) did not develop into structured 3D communities, while the majority of aggregates ($50 < P_0 < 300$) grew following the power law leading to microcolonies over time. These findings are in agreement with Kragh et al.'s study demonstrating enhanced fitness of aggregates (vs. single cells) under nutrient-rich conditions to form 3D biofilm structure although growth behaviors and structural organization can be also influenced by other factors such as metabolic activity and physical cell–cell contact[9,45]. A recent study using *Vibrio cholerae* biofilms and computational modeling showed that local cellular order and global biofilm architecture could arise from mechanical cell–cell interactions by regulating the production of specific matrix components[45].

Although the knowledge about the biology of bacterial cell growth on the surface and their biochemical events at a single-cell level has been increasingly described[2,20,39,46–48], details on the physical and biological interactions between communities when they are contacting and merging remain limited[49–52]. In our study, we observed that once the clusters developed, they further merge with each other (without competition), leading to the formation of biofilm superstructures. These findings are interesting as a previous study reported bacterial competition for space using *E. coli* biofilm model and computer simulation whereby

expansion (caused by different lag times of initial colonizers) and boundary-associated competition (caused by "pushing" interactions between growing microcolonies) were observed[53]. For *Bacillus subtilis*, the competition was confirmed for distant communities growing under flow, displaying an intriguing synchronized growth behavior, especially when there is a scarcity of nutrients[10]. For *S. mutans* biofilms, this synchronization behavior was not observed in the time range analyzed here. Rather, the individual microcolonies (distant or in close proximity) continued to grow without disruption until merging with each other, and the merged structures behaved and grew like a single new harmonized community. However, we observed that the structure and growth dynamics were influenced by inter-species interactions. Specifically, the presence of an antagonist (*S. oralis*) lowered the values for *a* and *b* coefficients, limiting the growth of *S. mutans* and the development of microcolonies in the mixed biofilm. Conversely, co-culturing with *C. albicans* (cooperative) resulted in enlarged mixed-species biofilms, without disrupting the bacterial growth dynamics reflecting a symbiotic behavior that promotes the spatial structuring and expansion as a whole.

These findings complement the previously observed short-range interactions at the single-cell level[5,9,54]. The dynamics of bacterial population growth and biofilm organization reported here suggest a communal behavior between microorganisms occurring across several spatial and temporal scales, which can be influenced by the type of neighboring cells (antagonistic or symbiotic), community merging and scaffolding. From a spatial and structural perspective, *S. mutans* biofilm development involves multiscale and multidimensional events as summarized in Fig. 8. (1) The bacterial cells (*S. mutans*) were able to bind to different surface topographies ("terrains") providing effective surface colonization ("settling capacity"), (2) a subset of the cell population can continue to grow, aggregate, and occupy the surface when structuring molecules are produced (e.g., exopolymeric substances—EPS), leading to organized and structured

communities to build 3D microcolonies ("buildings"). Conversely, without the EPS the bacterial cells are unable to aggregate and expand three-dimensionally. (3) These communities (microcolonies) can expand and merge with each other in a collaborative fashion, without competition between adjacent communities ("conurbation"), (4) competition (with *S. oralis*) or cooperation (with *C. albicans*) can impair the population growth or generate larger structures with distinct community organization and structural patterns. However, we emphasize the limitations of our urbanization analogy, which does not include anthropomorphic aspects of a human urban setting (such as traffic, mobility, governance, or supply lines).

The spatiotemporal biofilm development, from submicron to submillimeter-scale, described here also raises additional questions. While spatial distribution, inter-species and EPS interactions are key modulators of community expansion and merging, it is unclear how neighboring species coordinate or antagonize each other at the various length scales and dimensions[55,56]. Would specific signaling molecules[57,58] or other EPS such as proteins, eDNA, or β-glucan[21,59] influence on the coefficients in power law (a and b) or the scaffolding events? How the timing of the exposure of competing or symbiotic microbial species can alter the individual or communal behavior of the dynamic and static colonizers? In addition, cell behavior and their recovery after structural damage (by external fluid shear) or chemical insults (by antibiotics) at small and large scales simultaneously would bring new therapeutic perspectives to control biofilms. Therefore, it will be interesting to further assess the impact of these variables on the coefficients in power law and identify the modulating biomolecules driving the communal behavior.

In summary, our findings reveal additional biofilm-assembly mechanisms occurring at multiple scales. This process can be analyzed at large scales while considering the growth behavior of individual microorganisms colonizing on a surface (e.g., single cell, aggregates, or clusters) across different topographies using our analytical tool (BioSPA), which might be applicable to other biofilm systems, including those harboring polymicrobial communities. Further studies on the role of the individual cellular and matrix units at short-range scales and their impact on the communal behavior at large scales may advance the knowledge about the biofilm-assembly principles. Disruption of these highly interconnected events may help develop new therapeutic strategies to more effectively eradicate harmful biofilms associated with many infectious diseases and environmental biofouling.

## Methods

**Microorganisms and culture conditions**. *Streptococcus mutans* UA159 serotype c (ATCC 700610), a well-characterized biofilm-forming organism, or a green fluorescent protein (GFP)-expressing *S. mutans* strain (LDH-gfp; gift from Justin Merritt, Oregon Health & Science University) was used for single or mixed-species biofilm experiments with *Streptococcus oralis* or *Candida albicans*. *S. oralis* J22[60] (gift from Jens Kreth, Oregon Health & Science University) or *C. albicans* SN250 tagged with red fluorescent protein (tdTomato; gift from Damian J. Krysan, University of Iowa) strains were used for mixed-species biofilm experiments. The cultures were stored at −80 °C in tryptic soy broth (*S. mutans/S. oralis*) or Sabouraud dextrose broth (*C. albicans*) containing 20% glycerol. All strains were grown to mid-exponential phase (optical densities at 600 nm of 1.0 for *Streptococci* or 0.6 for *Candida*) in ultra-filtered (10-kDa molecular-mass cutoff membrane; Prep/Scale, Millipore, MA) buffered tryptone-yeast extract broth (UFTYE; 2.5% tryptone and 1.5% yeast extract, pH 7.0) containing 1% (wt/vol) glucose at 37 °C under 5% $CO_2$[37,61]. These bacterial suspensions were then prepared for inoculum for the biofilm growth experiments.

**Biofilm growth experiments**. Biofilms were formed using our saliva-coated hydroxyapatite (sHA) model[30,36]. Briefly, the sHA discs (surface area, 2.7 ± 0.2 cm²; Clarkson Chromatography Products, Inc., South Williamsport, PA) were placed in a vertical position using a custom-made wire disc holder, mimicking the free smooth

surfaces of the pellicle-coated teeth. Each disc was inoculated with microbial cells in UFTYE broth containing 0.1% (30 mM) sucrose at 37 °C under 5% $CO_2$ for 1 h, which promotes bacterial adhesion on the apatitic surface[30]. The bacterial suspensions with a defined microbial population of *S. mutans* ($10^8$ CFU/ml) were used as inoculum for the single-species biofilm experiments. For mixed-species biofilms, the suspensions were mixed to provide a defined population of *S. mutans* ($10^8$ CFU/ml) and *S. oralis* ($10^6$ CFU/ml) or *C. albicans* ($10^6$ CFU/ml, containing predominantly yeast cell forms), which are critical for the reproducibility of our model[61]. After 1 h of incubation, sHA discs harboring adhered cells were removed and inserted in the modified flow-cell microfluidics device (FC310, BioSurface Technologies, Bozeman, MT, USA) for real-time confocal imaging and computational analysis. All experiments denoted as "$t_0$" corresponds to the confocal scanning before starting the flow in the chamber displaced under the microscope objective lens (as detailed later). To assess the bacterial growth dynamics, we tested biofilm formation using different carbohydrates sources as follows: (i) high sucrose (1.0%; w/v), (ii) low sucrose (0.1%; w/v), (iii) feast and famine cycle (0.1% sucrose and 1% sucrose switched every hour), and (iv) glucose and fructose (0.5% each; w/v). Furthermore, we also examined the role of EPS on the bacterial growth dynamics and structural organization by using an EPS-degrading enzyme mixture that can effectively digest EPS glucan matrix without any antibacterial activity[31]. A pre-determined optimized enzyme mixture comprised mutanase (α-(1 → 3)-glucanase; EC3.2.1.59, gift from Johnson & Johnson) and dextranase (α-(1 → 6) glucanase; EC3.2.1.11, Sigma-Aldrich, St. Louis, MO, USA) (0.18 units each) were added to UFTYE medium supplemented with 1% sucrose. All time-lapsed biofilm experiments were done at a flow rate of 100 μL/min throughout the experimental period.

**High-resolution HAD surface topography and confocal imaging of biofilms**. The surface topography of the HAD surface (prior to biofilm formation) was analyzed by a nondestructive confocal contrasting method using Zeiss LSM 800 with a 40× (numerical aperture = 1.2) water immersion objective by detecting the reflected light of a 405-nm laser (window of detection from 420 to 450 nm) from a 3D scanning of the HAD. The images were processed using ConfoMap (Zeiss) to create 3D topography rendering and measure the surface properties in 3D (average roughness ($S_a$), root-mean-square roughness ($S_q$), and skewness ($S_{sk}$)). The dynamics of the biofilm development process on the HAD were assessed via a multi-labeling approach optimized for oral biofilm imaging[30,62] with some modifications using Zeiss LSM 800 with a 40× (numerical aperture = 1.2) water immersion objective. Briefly, *S. mutans* cells labeled with Syto 9 (485/498 nm; Molecular Probes Inc., Eugene, OR) or LDH-gfp *S. mutans* was excited using 488 nm laser, and bacterial fluorescence image was collected by a 480/40-nm emission filter (Fig. 1a, b). *S. oralis* labeled with SYTO 82 (541/560 nm; Molecular Probes) was excited using 561 nm laser, and fluorescence emitted was collected by a 560/40-nm emission filter. tdTomato-expressing *C. albicans* was excited using 561-nm laser, and the image was collected by a 560/40-nm emission filter. EPS labeled via incorporation of AlexaFluor 647 dextran conjugate (Molecular Probes) was excited using 640-nm laser, and fluorescence image was collected by a 660/40-nm emission filter. *S. mutans* and EPS were illuminated simultaneously, while *S. oralis* or *C. albicans* was illuminated sequentially to minimize crosstalk. The time-lapse scanning for bacteria, fungi, and EPS channels were acquired with a 0.42-μm step size with a 20 min interval. Amira 5.4.1 (Visage Imaging) and ImageJ were used to create 3D renderings to visualize the architecture of the biofilms.

**Image processing**. Raw image stacks from CLSM can be used for the 3D reconstruction of the surface, microorganisms and the EPS at each analyzed time point, thus resulting in 4D data. Each CLSM image stack can be converted to a scalar field, in which each pixel in the image stack is converted to a $(X, Y, Z)$ position. For determination of surface topography, a 3D-Gaussian filter with variance of 0.8 was applied to the raw stack (grayscale; 8-bit/channel) containing the light-reflection signal at $t_0$ using software ImageJ. After filtered, the stacks were converted into Numpy arrays (i.e., scalar fields) using Python Scikit-Image package[63] on Spyder environment, and the surface topography was determined by a script that scanned along the $Z$-axis all $X$–$Y$ positions of each stack, from the top to the bottom. During the top-to-bottom scanning of the CLSM image stack, it was collected the $(X, Y, Z)$ position that had the first detected grayscale contrast difference, thus resulting in a matrix with the HA disc topographical profile. Roughness parameters ($S_a$, $S_q$, and $S_{sk}$) were further calculated by following the equations:

$$Sa = \left(\tfrac{1}{S}\right) \int_0^S \left| h(x,y) - \overline{h(x,y)} \right| dA$$

$$Sq = \left[ \left(\tfrac{1}{S}\right) \int_0^S (h(x,y) - \overline{h(x,y)})^2 dA \right]^{1/2} .$$

$$Ssk = \left(\tfrac{1}{Sq^3}\right) \left(\tfrac{1}{S}\right) \int_0^S (h(x,y) - \overline{h(x,y)})^3 dA$$

where: $h(x,y)$ is the height profile; $S$ is the evaluated area.

In order to localize the microorganisms (bacteria and fungi) in the 3D image stacks, a 3D-Gaussian filter with variance 0.8 was applied to the raw stacks containing the fluorescence signal on ImageJ, and images were further binarized on the same software. As the contrast in the raw images was high due to the

SYTO9 high fluorescence, the binarization process was precise to reveal the presence or absence of microbes in the images. After converting the images to Numpy binary arrays, each white pixel (value 1) in the images stack was correlated to a $(X, Y, Z)$ position with the presence of the microorganism, where black pixels (value 0) to its absence. The segmentation processes performed in the binarized images stacks, such as to identify and isolate cells (single cells) and structured communities (i.e., chains, chain aggregates, clusters, microcolonies) was done through a 3D labeling algorithm[64] implemented in Python Scikit-Image package. After isolating each member of the population (see Fig. 1c), the cropped matrices were filtered to remove overlapped members, especially at the matrix edges. The determination of microorganism volume was then performed by summing (integrating) the Numpy binary arrays and multiplying by the pixel/volume ratio for each image stack (determined by CLSM conditions). Surface colonizers comprise single cells, aggregates, and clusters identified in image stacks obtained at $t_0$. Due to the small dimension of S. mutans (0.4–0.6 μm), population members were revealed at cell resolution just up the aggregates (see Fig. 1d). To estimate the cell number of aggregates and clusters at $t_0$, S. mutans single-cell volume was considered as $0.35\ \mu m^3$ (average value found). Microcolonies were treated as single elements with full connectivity (continuity) along the volume (see Fig. 1d), thus a cell number estimation was not performed for these communities. As the evolution of the surface colonizers toward structured communities occurs with EPS formation, beyond the cluster level (see Fig. 1d) there is not a linear relation between volume and cell number. For this reason, we used volume as the parameter to determine the bacterial growth ($V(t)$).

In addition, population morphometric analysis started by identifying all marginal positions of each identified member, which represent its surface. The convex Hull was generated from these positions related to their surface[34], and the volume of the Hull (qhull Volume) was then calculated. Regarding the EPS signal, although it was very precise to identify the EPS matrix from a morphological point of view, especially around the microbes aggregates and microcolonies, it was not used for volume quantification and morphometric analysis. This is because along with the increase in the signal from the EPS production and the community structuring process, there is also an increase in the signal along time from the adsorption of dextran-conjugated AlexaFluor-647 on the biofilm. This latter event introduces an artifact in the EPS volume determination, though it does not compromise morphological determination of the polymeric matrix. In this way, the EPS raw signal was used qualitatively without filtering and binarization process. All Z-projection images used in this study were calculated from the CLSM images stacks by collecting at each $(X, Y)$ position the most intense signal in the stack (i.e., max intensity Z-projection).

**Quantitative analysis**. Analysis of initial microbial attachment on HAD: In the surface topographical analysis, a comparison between occupied (by colonizers) and non-occupied surface sites was done by determining the normalized distribution of microbes as a function of topographical variables, such as $S_a$, $S_q$, and $S_{sk}$. On occupied sites, the determination of $S_a$, $S_q$, and $S_{sk}$ were done for the height values $h\ (x, y)$ of rectangular areas just under which colonizers were found (Fig. 1e). For non-occupied sites, the determination of $S_a$, $S_q$, and $S_{sk}$ values was done for multiple scales (multiscale area selection—MAS) by considering varied rectangular areas ($3–115\ \mu m^2$), collected from the whole-scanned area. $S_a$, $S_q$, and $S_{sk}$ equations are presented in the Supplementary Information. The cell number of surface colonizers was determined by considering an S. mutans cell volume of $0.35\ \mu m^3$, which is the mean value found from independent calculations (>100). Statistical variables mean, median, and quartiles for coefficients in fitted growth laws of S. mutans were determined for a population of more than 800 surface colonizers.

Population-growth analysis: We developed a simultaneous multiscale population growth and 3D-morphometric analyses to assess the spatiotemporal evolution of a large number (hundred-scale) of colonizers toward the formation of microcolony communities. To achieve this, we considered surface topography as the terrain available for colonization, microorganisms as individuals, and EPS as a scaffold, each separately. We were able to successfully evaluate all microorganisms (colonizers)initially attached to a surface and their surface-attachment sites (topographical information) from a 3D reconstruction (using Python Skimage and Numpy modules) across the entire confocal images data sets. Then, the biofilm development on the surface, including the EPS production from a single cell to individual microcolony level, was analyzed via 3D reconstruction of time-resolved CLSM image stacks (time-lapse experiment), resulting in 4D data. With time-resolved image stacks (4D data), each surface colonizer had its evolution tracked in terms of number, colony size and shape with the sequential use of 3D-Gaussian filter, image binarization process, elements labeling (i.e., microbes), elements convex hull and volumes determination, and fitting of the growth curves (i.e., volumes vs. time). This sequential processing was performed on each image stack of the 4D data. The Volume X Time growth curves were fitted with the power law $V(t) = a \cdot t^b$. The standard deviation for "a" and "b" coefficients calculated from the fitting of the growth curves had maximum values of $7.5 \times 10^{-4}$ and 0.15, respectively. The customized analytical toolbox (Biofilm Spatiotemporal Population Analysis; BioSPA) and a tutorial guide can be downloaded from https://github.com/

amaurijp/BioSPA, and the details of the protocols are provided in the Supplementary Information (Supplementary Fig. 1). In addition, BioSPA was validated with a mock biofilm dataset (see Supplementary Fig. 5).

**Statistical analyses**. One-way analysis of variance (ANOVA) with post hoc Tukey HSD test for multiple comparisons of biovolumes between S. mutans single-species and mixed-species biofilms was done. Analyses were performed with SPSS 19.0 (SPSS, Inc.). The level of statistical significance was set at 0.05.

**Reporting summary**. Further information on research design is available in the Nature Research Reporting Summary linked to this article.

## Data availability
The source data underlying Figs. 2, 3, 4, 5, 6, and 7 are provided as a Source Data file.

## Code availability
The BioSPA and a tutorial guide are readily available in GitHub (https://github.com/amaurijp/BioSPA), and can be downloaded for free.

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

## Acknowledgements

This work was supported in part by the National Institutes for Dental and Craniofacial Research (NIDCR) grants DE027970 (GH) and DE025220 (HK). A.J.P. is the recipient of the Brazilian agency CAPES scholarship grant (88881.119452/2016-01), and thank the support from Brazilian agencies CNPq and FUNCAP for grants PRONEX PR2-0101-00006.01.00/15 (Nucleus of Excellence on Physico-Chemistry at Extreme Conditions) and PRONEM PNE-0112-000480100/16.

## Author contributions

A.J.P., G.H., and H.K. conceived and designed the experiments. A.J.P. and G.H. collected and analyzed the data. A.J.P., G.H., and H.K. wrote the paper. All authors discussed the results and commented on the paper.

## Competing interests

The authors declare no competing interests.
