## [Peer Review File · Nature Communications]

Reviewers' comments:

Reviewer #1 (Remarks to the Author):

This manuscript reports a fascinating study of biofilm development. Using conventional confocal microscopy, the authors performed an unconventional analysis to obtain detailed information concerning how *Streptococcus mutans* cells developed from single cells through microcolonies towards a full biofilm. The findings are of sufficient general interest to be considered for publication in *Nature Communications*, but I do have two major and many minor issues with the manuscript as it stands. The authors need to address these issues and revise their manuscript before a final recommendation can be given.

My first major issue is to do with validation of the authors' methodology. The chief novelty consists of the claim that they can analyse their data meaningfully to obtain information from the single cell (micron) level up to the level of at least some significant portion of an entire biofilm. The details of the method are given partly in the methods section of the manuscript, and mostly in the supplement. This is all well and good. But I can see no evidence presented to validate the method. How do the authors know that they've got it right? As it stands, the manuscript asks the reader to take the authors' word for it. This won't do. Evidence must be given to verify that the authors' method of analysis indeed yields what they claim it yields.

My second major issue is that the authors do not interact with significant recent work along similar lines. Most importantly, Kragh et al. *mBio* 7, e00237-16 (2016) have studied the relative fate of single cells and pre-formed aggregates in the development of *Pseudomonas* biofilm development. They find that whether single cells or aggregates are more likely to grow into biofilms depends on nutrient conditions. It seems highly probable that this work has significant importance for interpreting the authors' findings here. In particular, the authors need to tell us whether the scenario they present is only valid under conditions of nutrient limitation and therefore consistent with Kragh et al. Without such clarification, we cannot assess the generality of the findings presented here.

Next, I list the more minor comments and issues in the order of their appearance in the manuscript.

Abstract – should mention the main species studied, viz., *S. mutans*.

p. 5, line 118: should it not be root mean squared roughness?

Throughout: S_a , S_q , S_{sk} are inconsistently subscripted or not – they should all be subscripted, including when used to label axes in the figures

p. 7 Figure 1e – the topography scale is given to two decimal places. I find it hard to believe that the surface features can be resolved to 10 nm using optical microscopy!

Figure 1f – I simply do not understand these graphs, because I have no idea what it means to plot these features against 'area'. What area? Why? Is this related to the square labelled with L in Figure 1e, which is not explained?

Figure 1g – what is the meaning of the blue arrow above the colour scale bar?

p. 8 line 177 *et passim* – I object to the notation V_{420} . Since the function $V(t)$ has already been defined, the value of this function at 420 minutes should be written as $V(420)$; there is absolutely no reason to use a subscript.

p. 8 line 179 – I see no evidence for ‘three distinct stages’ of growth. Either remove ‘distinct’ or give evidence for the growth stages being distinct, i.e. tell us how we demarcate one stage from another unambiguously. In any case, what were the ‘further analyses’?

p. 9 Figure 2c legend – what is the meaning of ‘flow’ for the pink bar? Does this mean ‘the start of flow’, i.e. t_0 ? Please clarify unambiguously.

Figure 2d – legend should refer not to gray and red, but to ‘data points’ and ‘continuous line’; the gray data points are far too hard to see. Also, to help the reader decide how good the power law fit is, please replot this as a log-log plot so that the power law part shows up as a straight line.

p. 10 lines 210-211 – I believe the authors mean ‘For actively growing cells (DC) with $P_0 \geq 75$, we found ...’

p. 10 lines 216-217 – ‘... whereas at a certain threshold ($P_0 > 50$ cells) ...’ The threshold has been 75 throughout this paragraph, why has it suddenly become 50? I see no threshold at 50 in any of the plots.

p. 11 community merging section – these findings in 3 dimensions should be discussed in the light of recent work, e.g., Lloyd and Allen, *J. R. Soc. Interface* **12**, 20150608 (2015), who studied the merging of bacterial colonies in 2 dimensions.

p. 18 Figure 5a, b – again plotting as log-linear should help the reader judge the quality of the power law fits.

Reviewer #2 (Remarks to the Author):

The authors present data that illustrates how surface structure, community structure, biofilm matrix composition, and time influence *S. mutans* biofilm development and dynamics. They present an interesting concept of comparing biofilm development to urbanization concepts. While these results are interesting and would be of benefit to the community, the paper can be improved. This paper currently suffers from lack of flow and clarity that could be greatly improved by explaining key concepts (power laws, urbanization concepts) and continuing the urbanization metaphor throughout. The figures could be improved by addressing color scheme issues and insufficient labeling.

Major concerns

- The flow of the introduction is confusing and redundant in places. For instance, the connection between bacteria growth power laws and urbanization concepts is hinted at, but never clearly explained. Giving a clear explanation on growth laws and how they are applied to this system as well as clearly outlining the connection to large scale urbanization would greatly improve this paper.
- The title of the paper suggests a stronger connection to using urbanization as a model for understanding biofilm formation. While this is a thread that appears somewhat randomly in the text, it should be more unifying (appearing frequently), as the title suggests.
- The figures should be reformatted to make them easier to interpret.
 - o The heat maps in Figs 1E, 1G, and 2B are hard to read/ interpret. Especially differentiating black and blue on 1E and shades of red, pink, orange in 2B. These could be improved by using a gradient with more colors.
 - o Lines 132, 160-164 refer to left and right panels in Fig 1F, there are 3 panels. This could be clarified by just labeling above each panel. The colors of these panels are also difficult to differentiate in some of the plots.
 - o Figure 4 A,B,D: It is unclear if the timeline applies to all of these panels or just A. If it just applies to A, can the authors please provide the times for B and D. This figure also needs scale bars.
 - o Figure 5C: Scale bars need to be added
- Creating a model that illustrates the metaphor between urbanization concepts and biofilm formation would be helpful. Drawing connections between early settlers, villages, cities, and colonizers, microcolonies, and aggregates would help inform the reader and allow the authors to stick closer to their metaphor throughout the paper without having to re-explain themselves (lines 63-66, 185-187). This model could even incorporate the cell number cutoffs of the different populations (lines 113-115) to help inform the reader.
- Line 177-179. It is never explained what a and b are—just refers to figure 2, which also does not explain what a and b are.
- Some comments in the results section would be more appropriate in the discussion (e.g. Lines 222-226, 274-276)
- Much of the discussion just repeats results is redundant.
-

Minor concerns

- The manuscript could be improved by thorough proofreading and grammar checking. Many sentences have subject/ verb agreement issues (i.e. Line 21, Line 24, line 215).
- The organisms used in this study should be mentioned in the abstract.
- Many of the sentences are convoluted and difficult to read and would benefit from rewording.
- There are methods that are detailed in inappropriate places and should be sequestered to the methods section (lines 74-79).
- The flow of some sentences is thrown off by unnecessary parenthetical phrases (e.g. lines 25, 27, 98, 99, 100, 101). Also, many abbreviations are introduced multiple times, instead of just introducing once and then using the abbreviation throughout the paper (e.g. C, MAS, n.c.a., P0)
- Line 67- simultaneously is unnecessary
- Line 112-115 is confusing. Maybe move Line 74-76 immediately before it.

- Line 176- "that" is unnecessary
- Line 286- dynamic colonizers should be DC
- Line 319- the difference in convex hull volume and total volume is unclear.
- Line 351- change it to "locally affect"
- Fig 5c add scale bars
- Line 500- Insert a space after the period

Reviewer #3 (Remarks to the Author):

Review of "Spatiotemporal dynamics of bacterial growth mimics large scale urbanization during biofilm development"

In this manuscript, Paula et al. characterize biofilm formation of *Streptococcus mutans* in detail, using computational image analysis. The authors follow biofilm growth from single cells that have attached to surfaces up to large-scale biofilms on the scale of several hundred microns. The results are primarily descriptive and do not provide insights into new mechanisms. Nevertheless, a quantitative characterization of biofilm growth of *Streptococcus mutans* is new in the literature, to my knowledge, and is the basis for further analysis.

Focusing on the results content, there are several interesting new findings in this manuscript that I have not previously seen in the literature. The most prominent ones are:

- The scale-spanning growth characterization of *S. mutans* biofilms with quantitative detail.
- Only 40% of the *S. mutans* surface-attached cells (the authors term them "dynamic colonizers") grow into communities, while 60% stay dormant.
- Neighboring colonies of *S. mutans* do not inhibit each other's growth rate during and after merging in both nutrient-rich and nutrient-poor conditions.
- Even though the symbiotic interaction between *S. mutans* and *C. albicans* is known, and the antagonistic interaction between *S. mutans* and *S. oralis* is also known, the authors demonstrate that these known interactions also occur in biofilm growth conditions and affect the spatial organization.

There are also major issues with this manuscript, which are listed below.

Major issues with the manuscript:

1. The authors make an analogy between *S. mutans* biofilm growth and urbanization of human populations. Superficially, this analogy appears to make sense, as biofilm colonies increase in size and incorporate other colonies, analogous to human cities that incorporate neighboring villages/cities. However, most spatially localized population growth satisfies this superficial analogy.

I am generally critical of this analogy between biofilm growth and human urbanization, because this analogy breaks down beyond the superficial concept of growth: Biofilms are generally not organized similarly to cities, because key features of cities are not present in biofilms (traffic, individual mobility, governance, supply lines). The authors claim that the analogy to urbanization "suggests communal behavior across large spatial scale" for biofilms (abstract, line 39, and other parts of the manuscript). I strongly disagree with this statement. The analogy is just an analogy and because of its superficial character, nothing can be concluded from the analogy. Suggestions derived from this analogy are wild guesses. These considerations make me question the purpose of raising this analogy in the manuscript, which seems to suggest no new insights into biofilms, but instead the analogy could mislead readers with an anthropomorphic view of bacterial communities.

If the authors would like to put their findings for *S. mutans* into a broader context beyond

bacteria, the authors could perhaps think about spatial population growth.

2. Line 122-124: The authors appear to be unaware of a number of papers on the effect of surface topography on bacterial attachment. I encourage the authors to search the web for papers relating to bacterial attachment and surface topography, which will quickly reveal a review on this topic (<https://avs.scitation.org/doi/10.1116/1.5054057>) and many more papers.

On the same topic: Please also note that line 26-28 of the abstract does not hold up given the previous literature. The authors have discovered that *S. mutans* settles randomly on the surface irrespective of surface topography, but certainly this discovery cannot be claimed for "bacteria" in general.

3. Growth law in Line 177 and following: Why did the authors use the equation $V(t) = V_0 + a t(b)$ rather than a standard exponential growth equation? From the data in Figure 2d and Figure 3d, 3e it seems that an exponential growth equation would have also fitted the data. At a basic level, growth is due to cell doubling, which is generally an exponential process. How would the authors explain a growth mechanism that leads to a $t(b)$ power law growth?

4. Line 399 discussion (also relating to the major issue 2 above): Why do the colonies merge in a "coordinated" manner? They appear to simply grow into each other. There does not seem to be any coordination. A null hypothesis for the observations would be that the *S. mutans* microcolonies simply grow into each other by simply trying to maximize their growth. This does not require "communal behavior across large spatial scales" (line 39) or "coordination".

Minor issues with the manuscript:

5. Line 21-22 first sentence of abstract: This sentence seems to have grammatical mistakes.

6. Line 22-24, second sentence of abstract: This sentence is unclear in terms of content. The authors should note that there has been a lot of literature on how single cells grow into microcolonies. Please also check the grammar of this sentence.

7. Please check the grammar throughout the manuscript, I stopped commenting on this after the abstract.

8. Line 50: "tridimensional"?

9. Line 114 and following: The terminology definition is not exactly clear. Apparently P_0 is the number of bacterial cells in a colonization unit after the 60 mins of initial incubation? But "Colonizers with $1 \leq P_0 \leq 5$ " appear to be single cells... so it is unclear to me what P_0 is exactly.

10. Line 177: please define V_{420} properly.

11. Figures: the font size in the figures is too small.

Reviewer: 1

This manuscript reports a fascinating study of biofilm development. Using conventional confocal microscopy, the authors performed an unconventional analysis to obtain detailed information concerning how *Streptococcus mutans* cells developed from single cells through microcolonies towards a full biofilm. The findings are of sufficient general interest to be considered for publication in Nature Communications, but I do have two major and many minor issues with the manuscript as it stands. The authors need to address these issues and revise their manuscript before a final recommendation can be given.

Answer:

We thank the reviewer for the positive remarks. We addressed all the issues raised by the reviewer point by point and included these points in the revised manuscript.

My first major issue is to do with validation of the authors' methodology. The chief novelty consists of the claim that they can analyze their data meaningfully to obtain information from the single cell (micron) level up to the level of at least some significant portion of an entire biofilm. The details of the method are given partly in the methods section of the manuscript, and mostly in the supplement. This is all well and good. But I can see no evidence presented to validate the method. How do the authors know that they've got it right? As it stands, the manuscript asks the reader to take the authors' word for it. This won't do. Evidence must be given to verify that the authors' method of analysis indeed yields what they claim it yields.

Answer:

We have uploaded our biofilm analytical toolbox (BioSPA) software and a tutorial guide at <https://github.com/amauriip/BioSPA>, which can be readily downloaded for free. In addition, we added a full coding with a detailed explanation in the supplemental data that can be verified by reviewers and readers, as suggested by the editor and the reviewer. Furthermore, we also validated the BioSPA collection and fitting performance using a mock dataset containing biofilm objects (TestSet; see on <https://github.com/amauriip/BioSPA> and Supplementary Figure S5). Briefly, in the TestSet, we generated a population of 6 colonizers growing by different growth laws. First, five colonizers were generated with a growth law where b exponent is 2.80 and the standard deviation (STD) of the fitting is 0.06 to mimic Dynamic Colonizers. Among these five, one colonizer was generated at the edge of the image stack to confirm the algorithm cropping routine: colonizers located at the edge are excluded from the population analysis. The standard deviation of 0.06 was used to confirm the algorithm fitting-cropping routine: colonizers not following power-law (STD > 0.1) are excluded from the population analysis and they are considered as "non-fitted" (see Figure 4b). Finally, one colonizer was generated with a growth law where b exponent is 1.0 (STD of the fitting is 0) to confirm perfect fitting. All colonizers started with an initial volume of $1 \mu\text{m}^3$ (representing 1 cell).

My second major issue is that the authors do not interact with significant recent work along similar lines. Most importantly, Kragh et al. mBio 7, e00237-16 (2016) have studied the relative fate of single cells and pre-formed aggregates in the development of *Pseudomonas* biofilm development. They find that whether single cells or aggregates are more likely to grow into biofilms depends on nutrient conditions. It seems highly probable that this work has significant importance for interpreting the

authors' findings here. In particular, the authors need to tell us whether the scenario they present is only valid under conditions of nutrient limitation and therefore consistent with Kragh et al. Without such clarification, we cannot assess the generality of the findings presented here.

Answer:

We apologize for this oversight. We have now included Kragh et al. findings in the discussion. Kragh et al. reported that the bacterial aggregates of *Pseudomonas aeruginosa* were more likely to grow into biofilm due to their enhanced fitness. The authors found that aggregates are more competitive than single cells in terms of growing into structured biofilms in a nutrient-rich environment because of their larger access to nutrients along the z-axis (i.e. aggregates are “taller” than single cells). In our study, *S. mutans* cells were bound to sHA surface as single-cells and as small aggregates after 60 min bacterial binding period. Interestingly, we found that most of the initial single-cells colonizers ($P_0 < 5$) did not develop into structured 3D communities, while the majority of aggregates ($50 < P_0 < 300$) grew following the power-law leading to microcolonies over time. These findings are in agreement with Kragh et al.'s study demonstrating enhanced fitness of aggregates (vs. single cells) under nutrient-rich conditions to form a 3D biofilm structure.

Next, I list the more minor comments and issues in the order of their appearance in the manuscript. Abstract – should mention the main species studied, viz., *S. mutans*.

Answer:

We added the name of species in the abstract.

p. 5, line 118: should it not be root mean squared roughness?

Answer:

Sq denotes root mean square roughness. We revised this accordingly.

Throughout: S_a , S_q , S_{sk} are inconsistently subscripted or not – they should all be subscripted, including when used to label axes in the figures

Answer:

We apologize for this. We made all the surface parameters consistently throughout the surface.

p. 7 Figure 1e – the topography scale is given to two decimal places. I find it hard to believe that the surface features can be resolved to 10 nm using optical microscopy!

Answer:

We agree with the reviewer that the topography scale should be one decimal place based on z-resolution under our imaging settings. Although the color scale in Figure 1e displays two digits decimal places, the actual values of S_a , S_q , S_{sk} reported here contain only one digit decimal as shown in Figure 1f. We fixed the scale of Figure 1e as indicated by the reviewer. We also split Figure 1 into two to improve readability, so Figure 1e is now Figure 2a.

Figure 1f – I simply do not understand these graphs, because I have no idea what it means to plot these features against ‘area’. What area? Why? Is this related to the square labelled with ΔL in Figure 1e, which is not explained?

Answer:

We have revised to clarify this issue. We assessed the influence of surface parameters on the initial bacterial attachment to saliva-coated HAD surface. To address this, we locally analyzed the surface topography of the sites upon which bacterial colonizers were bound. This was performed by cropping the image stack containing the topographical profile (i.e. heightmap) precisely at the regions where bacterial colonizers were found at t_0 and by calculating the surface roughness parameters S_a , S_q and S_{sk} for these sites. Furthermore, we also measured S_a , S_q and S_{sk} for the entire surface profile by splitting the heightmap ($319.57 \mu\text{m} \times 319.57 \mu\text{m}$) into squared areas of $\Delta L \times \Delta L$ (3 to $115 \mu\text{m}^2$), termed multiscale area mapping (MAS), to calculate the roughness of all possible binding sites on the surface, in all possible scales (Figure 2a). Figure 2b-g shows the size distribution of initial colonizers vs localized surface roughness parameters, showing that bacteria colonize the surface in a random fashion. We added an additional explanation to clarify this in the text. Please see lines 120-127 on page 6.

Figure 1g – what is the meaning of the blue arrow above the colour scale bar?

Answer:

Blue arrow in Figure 1g indicates the time point (30 min) where the bacterial detachment by flow dropped significantly. To avoid confusion, we removed the blue arrow and reformatted the figure using the standard 'hot' scale color scheme. Please see Figure 2h (previous Figure 1g).

p. 8 line 177 et passim – I object to the notation V_{420} . Since the function $V(t)$ has already been defined, the value of this function at 420 minutes should be written as $V(420)$; there is absolutely no reason to use a subscript.

Answer:

We agree with the reviewer. We revised the notation throughout the manuscript as the reviewer suggested.

p. 8 line 179 – I see no evidence for 'three distinct stages' of growth. Either remove 'distinct' or give evidence for the growth stages being distinct, i.e. tell us how we demarcate one stage from another unambiguously. In any case, what were the 'further analyses'?

Answer:

We classified biofilm developmental stages as follows: (i) initial colonization; (ii) individual development; and (iii) mutual development. We agree with the reviewer that the differences between stages (ii) and (iii) are not clear-cut based on bacterial growth in terms of power-law. This is a qualitative classification based on spatial and morphological assessments with cell numbers. Thus, we removed 'distinct' and 'growth' and clarified the sentences to explain the basis of this classification. Instead, we compared the number of cells of initial colonizers (single cells, clusters, and aggregates) and their biovolumes at 0 min and at the end of the experimental period at 420 min (number of initial colonizers vs biovolume at 420 min). As shown in Figure 4a (three colored-lines), at 420 min, most of the (1) single cells had less than the volume of $10 \mu\text{m}^3$, while (2) clusters mainly developed to a structure with a range of 10^2 to $10^3 \mu\text{m}^3$ and (3) aggregates became large microcolonies ($\sim 10^3 \mu\text{m}^3$). Please refer lines 202-219 on pages 11-12.

p. 9 Figure 2c legend – what is the meaning of ‘flow’ for the pink bar? Does this mean ‘the start of flow’, i.e. t_0 ? Please clarify unambiguously.

Answer:

“Flow” indicates the start of the flow in the chamber. We added this information in the Figure 3 caption to clarify this.

Figure 2d – legend should refer not to gray and red, but to ‘data points’ and ‘continuous line’; the gray data points are far too hard to see. Also, to help the reader decide how good the power law fit is, please replot this as a log-log plot so that the power law part shows up as a straight line.

Answer:

We thank the reviewer for pointing this out. We replotted Figure 2d using a log scale and increased the size of data points. Please check Figure 3d.

p. 10 lines 210-211 – I believe the authors mean ‘For actively growing cells (DC) with $P_0 > 75$, we found ...’

Answer:

We revised this sentence to clarify it.

p. 10 lines 216-217 – ‘... whereas at a certain threshold ($P_0 > 50$ cells) ...’ The threshold has been 75 throughout this paragraph, why has it suddenly become 50? I see no threshold at 50 in any of the plots.

Answer:

We apologize for this. During the analysis, we found that setting the threshold of aggregates as 50 is more appropriate than 75. However, we forgot to fix it in the paragraph. We revised the threshold for this paragraph and made it consistent throughout the manuscript.

p. 11 community merging section – these findings in 3 dimensions should be discussed in the light of recent work, e.g., Lloyd and Allen, J. R. Soc. Interface 12, 20150608 (2015), who studied the merging of bacterial colonies in 2 dimensions.

Answer:

We thank the reviewer for this suggestion. We have added the following sentences in the main text:

“A previous study reported bacterial competition for space using E. coli biofilm model and computer simulation whereby expansion (caused by different lag times of initial colonizers) and boundary-associated competition (caused by ‘pushing’ interactions between growing microcolonies) were observed.³⁵ Interestingly, however, we did not find significant alteration of the growth rate for the microcolonies during merging events under nutrient-rich (1% sucrose) condition (colored vertical lines in Figures 5d and 5e indicate the merging moment).”

p. 18 Figure 5a, b – again plotting as log-linear should help the reader judge the quality of the power law fits.

Answer:

We replotted the graph as suggested by the reviewer.

Reviewer: 2

The authors present data that illustrates how surface structure, community structure, biofilm matrix composition, and time influence *S. mutans* biofilm development and dynamics. They present an interesting concept of comparing biofilm development to urbanization concepts. While these results are interesting and would be of benefit to the community, the paper can be improved. This paper currently suffers from lack of flow and clarity that could be greatly improved by explaining key concepts (power laws, urbanization concepts) and continuing the urbanization metaphor throughout. The figures could be improved by addressing color scheme issues and insufficient labeling.

Answer:

We thank the reviewer for the positive comment, and have addressed each of the issues point-by-point and included them in the revised manuscript.

Major concerns

- The flow of the introduction is confusing and redundant in places. For instance, the connection between bacteria growth power laws and urbanization concepts is hinted at, but never clearly explained. Giving a clear explanation on growth laws and how they are applied to this system as well as clearly outlining the connection to large scale urbanization would greatly improve this paper.

Answer:

We thank the reviewer for this suggestion. We have revised the introduction to address the issues regarding conceptual clarity and redundancy. We also clarified that the biofilm development resembles the urbanization process based on spatial and structural organization, particularly at large length scales. In addition, we also included a new figure to illustrate the connection between spatial population growth and biofilm structuring across space and time with urbanization concept.

- The title of the paper suggests a stronger connection to using urbanization as a model for understanding biofilm formation. While this is a thread that appears somewhat randomly in the text, it should be more unifying (appearing frequently), as the title suggests.

Answer:

We have revised the title of the paper to indicate that the spatiotemporal dynamics of bacterial population growth resemble spatial and structural aspects of urbanization during biofilm development. We also revised the concept of connecting spatial population growth and biofilm structuring to urbanization throughout the manuscript, including a new schematic figure depicting the conceptual framework.

- The figures should be reformatted to make them easier to interpret.
 - o The heat maps in Figs 1E, 1G, and 2B are hard to read/ interpret. Especially differentiating black and blue on 1E and shades of red, pink, orange in 2B. These could be improved by using a gradient with more colors.

Answer:

We revised all the figures throughout to enhance clarity, including splitting into new figures with larger fonts and images. Previous Figure 1 was split to new Figures 1 and 2, and the previous Figure 2 was split to new Figures 3 and 4. All the heat map colors were also changed to the standard 'hot' scale color scheme.

o Lines 132, 160-164 refer to left and right panels in Fig 1F, there are 3 panels. This could be clarified by just labeling above each panel. The colors of these panels are also difficult to differentiate in some of the plots.

Answer:

Letters were added to refer to each panel.

o Figure 4 A,B,D: It is unclear if the timeline applies to all of these panels or just A. If it just applies to A, can the authors please provide the times for B and D. This figure also needs scale bars.

Answer:

The timeline was removed; instead, we added the time point that was acquired for each micrograph.

o Figure 5C: Scale bars need to be added

Answer:

We added a scale bar. Please see new Figure 7.

- Creating a model that illustrates the metaphor between urbanization concepts and biofilm formation would be helpful. Drawing connections between early settlers, villages, cities, and colonizers, microcolonies, and aggregates would help inform the reader and allow the authors to stick closer to their metaphor throughout the paper without having to re-explain themselves (lines 63-66, 185-187). This model could even incorporate the cell number cutoffs of the different populations (lines 113-115) to help inform the reader.

Answer:

We agree and have created a schematic diagram to clarify this metaphor. Please see Figure 8.

- Line 177-179. It is never explained what a and b are—just refers to figure 2, which also does not explain what a and b are.

Answer:

We clarified what the constants ‘a’ and ‘b’ are, as requested by the referee. Please see lines 197-200 on page 11.

- Some comments in the results section would be more appropriate in the discussion (e.g. Lines 222-226, 274-276)

- Much of the discussion just repeats results is redundant.

Answer:

We relocated those sentences to the discussion as the reviewer suggested. We revised the discussion to avoid redundancy.

Minor concerns

- The manuscript could be improved by thorough proofreading and grammar checking. Many sentences have subject/ verb agreement issues (i.e. Line 21, Line 24, line 215).

- The organisms used in this study should be mentioned in the abstract.

- Many of the sentences are convoluted and difficult to read and would benefit from rewording.

- There are methods that are detailed in inappropriate places and should be sequestered to the methods section (lines 74-79).
- The flow of some sentences is thrown off by unnecessary parenthetical phrases (e.g. lines 25, 27, 98, 99, 100, 101). Also, many abbreviations are introduced multiple times, instead of just introducing once and then using the abbreviation throughout the paper (e.g. C, MAS, n.c.a., P0)
- Line 67- simultaneously is unnecessary
- Line 112-115 is confusing. Maybe move Line 74-76 immediately before it.
- Line 176- “that” is unnecessary
- Line 286- dynamic colonizers should be DC
- Line 319- the difference in convex hull volume and total volume is unclear.
- Line 351- change it to “locally affect”
- Fig 5c add scale bars
- Line 500- Insert a space after the period

Answer:

We revised the manuscript thoroughly to address all these issues raised by the reviewer.

Reviewer: 3

Review of “Spatiotemporal dynamics of bacterial growth large scale during biofilm development”

In this manuscript, Paula et al. characterize biofilm formation of *Streptococcus* mutants in detail, using computational image analysis. The authors follow biofilm growth from single cells that have attached to surfaces up to large-scale biofilms on the scale of several hundred microns. The results are primarily descriptive and do not provide insights into new mechanisms. Nevertheless, a quantitative characterization of biofilm growth of *Streptococcus* mutants is new in the literature, to my knowledge, and is the basis for further analysis. Focusing on the results content, there are several interesting new findings in this manuscript that I have not previously seen in the literature.

Answer:

We thank the reviewer for the positive comments.

There are also major issues with this manuscript, which are listed below.

Major issues with the manuscript:

1. The authors make an analogy between *S. mutans* biofilm growth and urbanization of human populations. Superficially, this analogy appears to make sense, as biofilm colonies increase in size and incorporate other colonies, analogous to human cities that incorporate neighboring villages/cities. However, most spatially localized population growth satisfies this superficial analogy.

I am generally critical of this analogy between biofilm growth and human urbanization, because this analogy breaks down beyond the superficial concept of growth: Biofilms are generally not organized similarly to cities, because key features of cities are not present in biofilms (traffic, individual mobility, governance, supply lines). The authors claim that the analogy to urbanization “suggests communal behavior across large spatial scale” for biofilms (abstract, line 39, and other parts of the manuscript). I strongly disagree with this statement. The analogy is just an analogy and because of its superficial character, nothing can be concluded from the analogy. Suggestions derived from this analogy are wild guesses. These considerations make me question the purpose of raising this analogy in the manuscript, which seems to suggest no new insights into biofilms, but instead the analogy could mislead readers with an anthropomorphic view of bacterial communities.

If the authors would like to put their findings for *S. mutans* into a broader context beyond bacteria, the authors could perhaps think about spatial population growth.

Answer:

We agree with the referee that an improper analogy could lead readers to have an anthropomorphic view or human urban life-style of the biofilm development process, which was not our intention. We also agree that this point could also cause confusion about our analogy suggesting communal behavior across a large scale. We also clarified that the population growth (i.e. increase in the number of bacterial cells) by itself is not the reason for the urbanization analogy.

The urbanization analogy is related to both spatial and structural aspects of population growth and expansion, particularly at large length scales whereby multiple population growth and merging events, from hundreds of single cells and clusters to microcolonies, at different locations on the surface were analyzed simultaneously. Although general analogies have been made, visualization and

quantification of the spatial growth of individual cells, neighboring clusters and colonies and their merging events and spatial structuring across multiple locations, time and length scales have not been, to the best of our knowledge, determined. However, we realize that our conceptual framework lacked both clarity and details to connect biofilm development and urbanization from spatial and structural perspectives.

We have thoroughly revised the manuscript to address these conceptual issues, while also clearly stating the limitations of this analogy in our discussion to avoid misunderstandings as raised correctly by the reviewer. In addition, we have changed the title and emphasized the resemblance of the structural aspects of the urbanization with biofilm development, including an additional figure with schematics to illustrate this point more clearly.

2. Line 122-124: The authors appear to be unaware of a number of papers on the effect of surface topography on bacterial attachment. I encourage the authors to search the web for papers relating to bacterial attachment and surface topography, which will quickly reveal a review on this topic (<https://avs.scitation.org/doi/10.1116/1.5054057>) and many more papers.

On the same topic: Please also note that line 26-28 of the abstract does not hold up given the previous literature. The authors have discovered that *S. mutans* settles randomly on the surface irrespective of surface topography, but certainly this discovery cannot be claimed for “bacteria” in general.

Answer:

We apologize for this oversight. We totally agree with the reviewer that we cannot make a general conclusion that all bacteria bind to surface randomly. We have emphasized that our findings are related to *S. mutans* binding to saliva-coated hydroxyapatite (sHA) disc surface under the experimental conditions employed here. We have clarified this point in the abstract and in the introduction

3. Growth law in Line 177 and following: Why did the authors use the equation $V(t) = V_0 + a \cdot t^b$ rather than a standard exponential growth equation? From the data in Figure 2d and Figure 3d, 3e it seems that an exponential growth equation would have also fitted the data. At a basic level, growth is due to cell doubling, which is generally an exponential process. How would the authors explain a growth mechanism that leads to a t^b power law growth?

Answer:

We thank the referee for the comment. Indeed, the exponential growth of planktonic bacteria is due to cell doubling. However, during the biofilm formation, the growth can deviate significantly from the exponential law predicted for the planktonic state (growth rate can be even oscillatory in a nutrient-rich environment: Liu et al., Nature 523, 550-554, 2015). In our study, we found that equation $V(t) = V_0 + a \cdot t^b$ fits the best with the lowest error considering all curves analyzed, compared to the standard exponential growth equation. The bacterial population growth was successfully fitted with this equation. Interestingly, we also demonstrated that EPS production has a major contributing role to coefficient ‘a’ value in equation, but not ‘b’ value.

4. Line 399 discussion (also relating to the major issue 2 above): Why do the colonies merge in a “coordinated” manner? They appear to simply grow into each other. There does not seem to be any coordination. A null hypothesis for the observations would be that the *S. mutans* microcolonies simply grow into each other by simply trying to maximize their growth. This does not require “communal behavior across large spatial scales” (line 39) or “coordination”.

Answer:

We thank the reviewer for pointing this out. We have removed the term “coordination” and “communal behavior”. The main finding here is that there is no significant alteration of the growth rate of *S. mutans* colonies during merging events under a nutrient-rich condition. This is a unique feature of *S. mutans* during biofilm formation that allows colonies to merge harmonically (without competitive interference) to form a larger biofilm super-structure as a result of multiple merging colonies. We revised these points in the manuscript.

Minor issues with the manuscript:

5. Line 21-22 first sentence of abstract: This sentence seems to have grammatical mistakes.

6. Line 22-24, second sentence of abstract: This sentence is unclear in terms of content. The authors should note that there has been a lot of literature on how single cells grow into microcolonies. Please also check the grammar of this sentence.

7. Please check the grammar throughout the manuscript, I stopped commenting on this after the abstract.

8. Line 50: “tridimensional”?

Answer:

We thank the reviewer and have revised them accordingly.

9. Line 114 and following: The terminology definition is not exactly clear. Apparently P_0 is the number of bacterial cells in a colonization unit after the 60 mins of initial incubation? But “Colonizers with $1 \leq P_0 \leq 5$ ” appear to be single cells... so it is unclear to me what P_0 is exactly.

Answer:

We apologize for the confusion. Initial *S. mutans* binding to sHA was first performed for 60 min, and then *S. mutans* bound onto sHA disc was transferred to flow cell to observe biofilm dynamics. Thus, P_0 indicates the number of *S. mutans* bound to sHA disc surface after 1h binding. We classified the initial colonizer as single cells, clusters, and aggregates depending on the number of *S. mutans* cells found adsorbed on the sHA surface. We clarified this point in the revised text.

10. Line 177: please define V420 properly.

11. Figures: the font size in the figures is too small.

Answer:

We thank the reviewer and have revised them accordingly.

REVIEWERS' COMMENTS:

Reviewer #1 (Remarks to the Author):

[No further comments for author.]

Reviewer #3 (Remarks to the Author):

Summary of improvements:

The revised manuscript has been improved in a number of places:

- The abstract is now much better in terms of accuracy of content description, and in terms of placing this manuscript into the context of previous literature. The new title is also a better description of the content.
- The description and justification of the analogy between biofilm growth and urbanization is now improved in the introduction, results, and discussion.

Summary of evaluation:

I am still unconvinced of the usefulness of the urbanization analogy for the biofilm research community, because

- it is superficial and only applies to particular features of biofilm growth and urbanization,
- there are numerous differences between urbanization and biofilm growth, and
- it encourages an anthropomorphic view of biofilms which can be misleading.

These are criticisms I also raised in my first review. However, I think the manuscript is now accurate in terms of description of the content, results, and discussion.

I value differences in opinion because they enrich the discourse, and I will therefore not oppose publication.

Generally, the data on the biofilm formation process of *S. mutans* and the biofilm growth equation for *S. mutans* is very interesting and a good addition to the literature on biofilms.

A few more important requested changes:

- I now noticed that the experimental geometry in Fig 1a is not entirely clear: Are the hydroxyapatite disks placed inside the flow channel? Or are they part of the surface of the flow channel? A more precise representation (perhaps with annotations next to the drawing) would be very helpful in Fig 1a.
- In Fig 2 (a2), Fig 3a could the authors please indicate where the 3rd dimension in the plot is? Please also annotated the units of the numbers on the axes in Fig 3a.

Minor comments:

- Line 20, line 25: the word "evolve" should be avoided here, to avoid confusion with evolution. Perhaps "grow" or "develop" would be more appropriate.
- In the caption of Fig 4 (line 222 and following), and line 198: The authors use the nomenclature "V(420)" and "V(0)". Please make sure to give the units  $V(t = 420 \text{ min})$, or $V(420 \text{ min})$.
- There are a number of typos in the blue text. I guess the copyeditor will fix them, or the authors could do this themselves.
- Please note: "tridimensional"  "three-dimensional" (this typo is in several places throughout the manuscript).

REVIEWERS' COMMENTS:

Reviewer #1 (Remarks to the Author):

[No further comments for author.]

Reviewer #3 (Remarks to the Author):

Summary of improvements:

The revised manuscript has been improved in a number of places:

- The abstract is now much better in terms of accuracy of content description, and in terms of placing this manuscript into the context of previous literature. The new title is also a better description of the content.
- The description and justification of the analogy between biofilm growth and urbanization is now improved in the introduction, results, and discussion.

Summary of evaluation:

I am still unconvinced of the usefulness of the urbanization analogy for the biofilm research community, because

- it is superficial and only applies to particular features of biofilm growth and urbanization,
 - there are numerous differences between urbanization and biofilm growth, and
 - it encourages an anthropomorphic view of biofilms which can be misleading.
- These are criticisms I also raised in my first review. However, I think the manuscript is now accurate in terms of description of the content, results, and discussion.

I value differences in opinion because they enrich the discourse, and I will therefore not oppose publication.

Answer:

We thank the referee for the positive comments and constructive critique.

Generally, the data on the biofilm formation process of *S. mutans* and the biofilm growth equation for *S. mutans* is very interesting and a good addition to the literature on biofilms.

A few more important requested changes:

- I now noticed that the experimental geometry in Fig 1a is not entirely clear: Are the hydroxyapatite disks placed inside the flow channel? Or are they part of the surface of the flow channel? A more precise representation (perhaps with annotations next to the drawing) would be very helpful in Fig 1a.

Answer:

We add more information on this matter in the Figure legend. The hydroxyapatite disc is placed inside the flow cell in the same position as shown in the diagram (Fig 1b).

- In Fig 2 (a2), Fig 3a could the authors please indicate where the 3rd dimension in the plot is? Please also annotated the units of the numbers on the axes in Fig 3a.

Answer:

To clarify this point, we added the following sentence to the Figure legend:
“Bacterial cells are shown in blue in the top view of a 3D representation of the bacterial cells (a2). Z-axis is shown in red and represents a scale of 48 μm (a2).”

Minor comments:

- Line 20, line 25: the word “evolve” should be avoided here, to avoid confusion with evolution. Perhaps “grow” or “develop” would be more appropriate.

Answer:

Done.

- In the caption of Fig 4 (line 222 and following), and line 198: The authors use the nomenclature “ $V(420)$ ” and “ $V(0)$ ”. Please make sure to give the units  $V(t = 420 \text{ min})$, or $V(420 \text{ min})$.

Answer:

Done.

- There are a number of typos in the blue text. I guess the copyeditor will fix them, or the authors could do this themselves.

Answer:

Done.

- Please note: “tridimensional”  “three-dimensional” (this typo is in several places throughout the manuscript).

Answer:

Done.